# POLYSKILL: LEARNING GENERALIZABLE SKILLS THROUGH POLYMORPHIC ABSTRACTION FOR CONTINUAL LEARNING

**Simon Yu**[1]  **Gang Li**[2]  **Weiyan Shi**[†1]  **Peng Qi**[†2]

[1]Northeastern University  [2]Uniphore

[†] Co-Supervision

{yu.chi, we.shi}@northeastern.edu, {gang.li, peng.qi}@uniphore.com

## ABSTRACT

Large language models (LLMs) are moving beyond static uses and are now powering agents that learn during their interaction with external environments. For example, agents can learn reusable skills while navigating web pages or toggling new tools. However, existing methods for skill learning often create skills that are over-specialized to a single website and fail to generalize. We introduce PolySkill, a new framework that enables agents to learn generalizable and compositional skills. The core idea, inspired by polymorphism in software engineering, is to decouple a skill's abstract goal (*what* it accomplishes) and its concrete implementation (*how* it is executed). Experiments show that our method (1) improves skill reuse by 1.7x on seen websites and (2) boosts success rates by up to 9.4% on Mind2Web and 13.9% on unseen websites, while reducing steps by over 20%. (3) In self-exploration settings without specified tasks, our framework improves the quality of proposed tasks and enables agents to learn generalizable skills that work across different sites. By enabling the agent to identify and refine its own goals, the PolySkill enhance the agent a better curriculum, leading to the acquisition of more generalizable skills compared to baseline methods. This work provides a practical path toward building agents capable of continual learning in adaptive environments. Our findings show that separating a skill's goal from its execution is a crucial step toward developing autonomous agents that can learn and generalize across the open web continuously. Our code can be found in https://github.com/simonucl/PolySkill.

## 1 INTRODUCTION

LLMs have enabled significant progress in agent development (Yao et al., 2023b; Yang et al., 2024; Agashe et al., 2025). Web agents represent a key class of LLM-based agents, where the agents navigates complex Graphical User Interfaces (GUIs) to achieve user-defined goals (Deng et al., 2023; Zhou et al., 2024a; Cheng et al., 2024; Wu et al., 2024b; Chen et al., 2025b; Gou et al., 2025; Xue et al., 2025). However, a primary challenge lies in developing generalizable agents that can operate robustly across different tasks within a single website, as well as transfer skills to distinct websites within the same functional domain (e.g., generalizing across different airline booking interfaces). To be generalizable, these agents must learn from their experiences, allowing them to adapt when faced with new tasks or unseen websites (Silver & Sutton, 2025; Shen et al., 2025).

One promising direction is skill induction: learning reusable skills from past experiences. This approach was first explored in an open-ended environment by Voyager (Wang et al., 2023). Agent Workflow Memory (Wang et al., 2024) pioneered skill induction for web agents, which used natural language skills to prove the concept's viability. Consequently, Agent Skill Induction (Wang et al., 2025) and SkillWeaver (Zheng et al., 2025) made these skills more robust by structuring them as code. However, these methods primarily focus on *same website, cross-task* settings. By optimizing

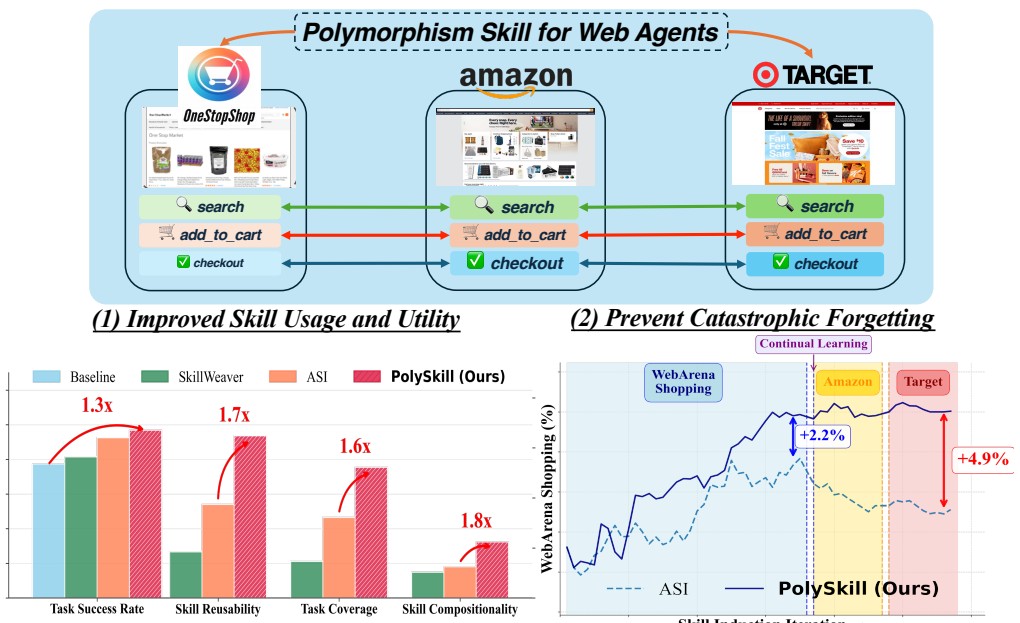

Figure 1: PolySkill, a novel approach that enables web agents to develop polymorphic skills that generalize across websites. PolySkill achieves superior performance with 1.3–1.8× improvements in task success rate, skill reusability, and compositional capabilities compared to existing methods.

for performance on familiar websites, they generate over-specialized skills that fail to generalize, leaving the critical challenge of cross-website generalization under-explored. This challenge highlights a fundamental tension between a skill's specificity and its generalizability. This leads us to two key research questions. First, *how can we induce skills that are transferable across diverse websites?* Second, *beyond task success, how can we quantitatively measure skill transfer and reuse?* Answering these questions is a crucial step toward the broader vision of agent autonomy, where an agent could discover such generalizable skills on its own through open-ended exploration (Liu et al., 2025a).

To address these questions, we introduce PolySkill, a framework that grounds agent skill learning in the principle of polymorphic abstraction, a cornerstone of object-oriented design from software engineering (Milner, 1978). This paradigm separates the abstraction from its actual implementations. We apply this to skills by (1) defining an abstract class (e.g., `AbstractShoppingSite`) that serves as a common interface for a domain, specifying high-level goals like the method "search(query, filter)". Concrete subclasses (e.g., `AmazonSite`, `TargetSite`) then provide the distinct, website-specific implementations. This allows the agent to operate at an abstract level and create compositional skills that are not tied to any specific website's functionality, decoupling them from brittle UI changes across websites. This approach also enhances better composition, allowing the agent to chain together abstract operations in the parent class like `search_product()`, `addToCart()`, and `checkout()` to execute complex, multi-step tasks; while preventing the need to implement the compositional skills per website.

To validate our approach, we address a critical gap in existing evaluation methods. Prior work on skill induction primarily relies on final *task success rates* (Zhou et al., 2024a). However, this metric reveals only part of the picture: it confirms *that* an agent succeeded, but not *how*. It cannot distinguish between success from efficiently reusing learned skills and success from solving a task from scratch, which makes it difficult to measure the value of the skill induction. To provide a clearer picture and answer our second research question, we introduce additional metrics, including *Skill Reusability* and *Task Coverage*, to directly diagnose the skill transfer failures that plague existing methods. Our evaluation reveals a clear improvement: while prior methods show a Skill Reusability below 18% on unseen websites, skills learned via PolySkill achieve a 31% reuse rate. Furthermore, we present the first comprehensive evaluation of skill induction on recent open-source agentic

models, such as Qwen3-Coder (Qwen, 2025) and GLM-4.5 (GLM, 2025), demonstrating that our findings are robust and not limited to proprietary models.

Finally, we extend our analysis to a task-free, continual learning setting (Zheng et al., 2025; Liu et al., 2025a), which tests an agent's ability to explore multiple websites and induce skills without predefined tasks. Our results suggest that this is a viable path toward self-improving agents, as the PolySkill better guides exploration than previous unstructured approaches (Zheng et al., 2025). More broadly, we believe the principle of polymorphic abstraction extends beyond the web, offering a promising direction for developing transferable skills for any agent that operates in diverse environments with shared structural patterns (Xu et al., 2025). By grounding skill creation in polymorphism, this work takes an important step toward continual learning, enabling agents to build a library of adaptive skills that can evolve with experience.

## 2 RELATED WORK

**Memory and Skill Acquisition for Agentic Learning** An agent's ability to generalize is fundamentally tied to how it represents skills. Current approaches often store skills in concrete formats, such as natural language descriptions in prompt libraries (Wang et al., 2024; Zhu et al., 2025) or brittle action traces from successful task executions (Wang et al., 2025). While more robust, programmatic representations, where skills are stored as executable code, also face limitations. These learned programs are typically concrete implementations tailored to a specific context (e.g., one website's UI), hindering their reuse across varied environments that serve a similar function (Wang et al., 2023; Zheng et al., 2025). These methods lack a mechanism for abstracting the semantic intent of a skill away from its specific implementation, which is crucial for flexible adaptation.

We address this gap with a novel, hybrid skill representation inspired by object-oriented design. While broader work has focused on the architecture of agent memory, such as using episodic streams (Park et al., 2023) or managed external stores (Chhikara et al., 2025; Fang et al., 2025), our contribution lies in the representation of the skill itself. We define a skill with an **abstract interface** that captures its semantic purpose, which can be linked to multiple, interchangeable **concrete implementations**. Drawing on established findings that abstraction and structure enhance agent reasoning and robustness (Wu et al., 2024a; Yao et al., 2023a), this polymorphic structure provides the formal benefits of programmatic skills while explicitly incorporating the flexibility required for generalization across different web pages.

**Continual Learning** A long-standing goal in AI is continual learning (van de Ven et al., 2025; Liu et al., 2025a), where an agent continually learn during testing time and its interaction with the environments. This is specifically true for web agents, where interaction is more effective than just thinking longer (Shen et al., 2025; Liu et al., 2025b). A key insight from AI research is that this requires **compositional generalization**, the ability to learn primitive concepts and recombine them to solve novel problems (Jiang et al., 2025). Existing web agents that learn from experience, such as those using exploration enhanced by curriculum learning (Zheng et al., 2025) or reinforcement learning (Zhou et al., 2024b), perform a simple form of continual learning by adding new skills to a library. However, their skills lack a compositional structure; a learned code snippet or action trace cannot be easily modified or combined with others in a principled way, leading to low utilization rates on new websites. Our work, first showing the effectiveness of the principles for modularity from *software engineering*, offers a powerful solution. Polymorphism (Milner, 1978) is a time-tested concept designed explicitly to manage variation between implementations while maintaining a stable interface. By applying this principle to agent skills, our approach provides a structured paradigm for learning capabilities that are modular and interchangeable. See Appendix B for detailed related work.

## 3 POLYSKILL: POLYMORPHISM-GUIDED AGENT SKILL INDUCTION

Our approach addresses the fundamental challenge of learning agent skills that balance specialization with generalization. We propose a hierarchical framework that separates skill learning into three complementary stages: skill discovery through polymorphic abstraction, skill refinement through compositional verification, and skill deployment through adaptive execution.

## 3.1 PRELIMINARY

**Problem Formulation.** We model the web agent's interaction environment as a Partially Observable Markov Decision Process (POMDP), defined by the tuple $\langle \mathcal{S}, \mathcal{A}_p, \mathcal{T}, \Omega, \mathcal{O} \rangle$. Here, $\mathcal{S}$ is the latent state space representing the full underlying state of the web application. $\mathcal{A}_p$ is the set of primitive actions the agent can execute on a webpage (e.g., `click(element)`, `type(text)`), as defined in Appendix E.2. The function $\mathcal{T} : \mathcal{S} \times \mathcal{A}_p \to \Delta(\mathcal{S})$ is the stochastic state transition function. Since the agent cannot perceive the entire state $\mathcal{S}$, it receives an observation $o_t \in \Omega$ (e.g., the A11y tree and viewport screenshot) at each timestep $t$ through the observation function $\mathcal{O} : \mathcal{S} \to \Delta(\Omega)$.

**LM-based Agent Policy.** We consider an agent driven by a large language model (LM) backbone, $\mathcal{L}$. The agent's policy, $\pi_{\mathcal{L}}$, determines the next action based on its current context. This context consists of a **working memory** $\mathcal{M}$, which stores the high-level task instruction and the history of observations and actions, and a dynamic **skill library** $\mathcal{K}_t$. The skill library contains reusable skills that expand the agent's full action space to $\mathcal{A}_t = \mathcal{A}_p \cup \mathcal{K}_t$. Each skill $k \in \mathcal{K}_t$ is a parameterized sequence of actions $k(\text{args}) := a_1 \oplus \cdots \oplus a_n$, where $\oplus$ denotes sequential execution and each action $a_i \in \mathcal{A}_t$, which includes both primitive actions ($\mathcal{A}_p$) and learnt skills ($\mathcal{K}_t$). The policy is thus denoted as $\pi_{\mathcal{L}}(a_t | o_t, \mathcal{M}_t, \mathcal{K}_t)$, which we shorten to $\pi_{\mathcal{L}}$.

**Task Execution and Objective.** The agent's goal is to complete a task specified by a natural language instruction $q$. At each timestep $t$, the agent receives an observation $o_t$, updates its memory $\mathcal{M}_t$, and selects an action $a_t \in \mathcal{A}_t$ using its policy. This interaction over a horizon $H$ generates a trajectory $\tau = (o_0, a_0, o_1, a_1, \ldots, o_{H-1}, a_{H-1})$. A task is considered successful if the trajectory satisfies a goal condition, indicated by a success function $g(\tau, q) = 1$. Our central objective is to induce an effective skill library $\mathcal{K}$. We formalize this by maximizing an efficiency-aware reward, $\max_{\pi_{\mathcal{L}}, \mathcal{K}} \mathbb{E}_{q \sim \mathcal{Q}}[g(\tau, q) - \gamma |\tau|]$, where the penalty on trajectory length $|\tau|$ incentivizes the creation of compact and reusable skills. While this objective could be optimized as a loss function, we instead use this efficiency principle to guide our agent's prompting.

The core challenge, which *PolySkill* addresses, is to populate $\mathcal{K}$ with skills that are both effective for specific contexts (specialized) and transferable to new tasks/domains (polymorphic).

## 3.2 THE POLYSKILL FRAMEWORK

**Limitation of existing skill induction methods** We identified the limitations of current skill induction methods. We tested two state-of-the-art approaches, ASI and SkillWeaver, on how well their learned skills transfer to unseen websites. Their example of skills can be seen in Table 4. As illustrated in Figure 2, our analysis revealed two key problems. First, the learning process can be unstable, producing **over-specialized skills**. For instance, SkillWeaver's performance with Claude-3.7-Sonnet degrades over time because its self-proposed tasks become increasingly complex and specific. This causes the resulting skills to be too intricate and poorly suited for generalization. Second, these skills show **poor generalization** when applied to new websites. This is reflected in extremely low Skill Reusabilitys on unseen websites: less than 9% for ASI and less than 3% for SkillWeaver.

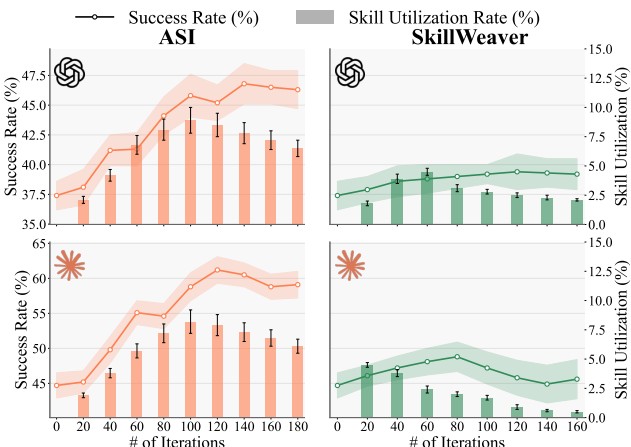

Figure 2: **Limitations of existing skill induction methods.** We evaluate ASI and SkillWeaver across two foundation models: 🍀 GPT-4.1 and ✹ Claude-3.7-Sonnet. Both methods show unstable learning dynamics and poor skill reusability, demonstrating over-specialization issues that hurt performance on WebArena Shopping tasks.

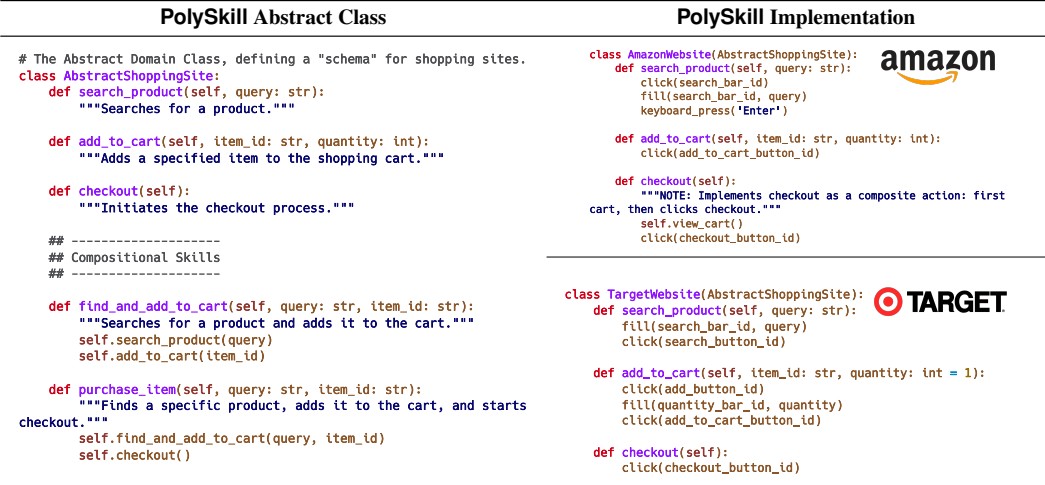

| PolySkill Abstract Class | PolySkill Implementation |
|---|---|

```python
# The Abstract Domain Class, defining a "schema" for shopping sites.
class AbstractShoppingSite:
    def search_product(self, query: str):
        """Searches for a product."""

    def add_to_cart(self, item_id: str, quantity: int):
        """Adds a specified item to the shopping cart."""

    def checkout(self):
        """Initiates the checkout process."""

    ## --------------------
    ## Compositional Skills
    ## --------------------

    def find_and_add_to_cart(self, query: str, item_id: str):
        """Searches for a product and adds it to the cart."""
        self.search_product(query)
        self.add_to_cart(item_id)

    def purchase_item(self, query: str, item_id: str):
        """Finds a specific product, adds it to the cart, and starts
checkout."""
        self.find_and_add_to_cart(query, item_id)
        self.checkout()
```

```python
class AmazonWebsite(AbstractShoppingSite):
    def search_product(self, query: str):
        click(search_bar_id)
        fill(search_bar_id, query)
        keyboard_press('Enter')

    def add_to_cart(self, item_id: str, quantity: int):
        click(add_to_cart_button_id)

    def checkout(self):
        """NOTE: Implements checkout as a composite action: first
cart, then clicks checkout."""
        self.view_cart()
        click(checkout_button_id)

class TargetWebsite(AbstractShoppingSite):
    def search_product(self, query: str):
        fill(search_bar_id, query)
        click(search_button_id)

    def add_to_cart(self, item_id: str, quantity: int = 1):
        click(add_button_id)
        fill(quantity_bar_id, quantity)
        click(add_to_cart_button_id)

    def checkout(self):
        click(checkout_button_id)
```

Table 1: Example of PolySkill. **(Left)** shows the high-level abstraction of the skills under shopping domains; **(Right)** shows the website-specific implementation across shopping domains, built upon the Abstract Shopping parent class. Note that the compositional skills would not need to be redefined, since it solely rely on the compositionality of other skills.

We build on prior work that demonstrates the robustness of representing skills as code (Wang et al., 2025; Liu et al., 2025a). To specifically address the brittleness and over-specialization, we introduce a solution inspired from software engineering: polymorphism. This allows our framework to separate a skill's abstract goal from its concrete, site-specific implementation.

However, these skills are often tied to one specific website's design, making them hard to reuse on other sites. Other methods, like SkillWeaver (Zheng et al., 2025), create robust skills by generating complex code from the agent's experience. As shown in Figure 2, these skills use very specific code to find elements on a page, limiting them to only one site. This over-specialization makes the skills brittle: they work well on the original website but break easily on new sites with different layouts.

**Our Proposed Solution** We introduce **PolySkill**, a framework that solves this problem by learning a domain-driven skill hierarchy. Instead of treating skills as isolated scripts, we organize them into classes based on a website's category. For example, skills for Amazon and Target are treated as concrete implementations of an abstract `AbstractShoppingSite` class. This structure allows the agent to learn a general "schema" for a type of website and then fill in the specific, reliable implementations for each new site it encounters.

## 3.3 SKILL INDUCTION PROCESS

**Base of Skill Induction Pipeline** Our skill induction process is built upon the robust verification pipeline established by ASI (Wang et al., 2025). In their framework, skill creation begins after a task is successfully completed using a sequence of primitive actions. An LLM-based induction module analyzes this successful trajectory to propose one or more programmatic skills that encapsulate reusable parts of the workflow (Pan et al., 2024). Before the skills are added to the library, a verification phase is done where the agent attempts to solve the same task again, this time by executing the newly generated skill. Only if this new execution is deemed as successful is the skill considered validated and added to the agent's library for future use.

**Innovation via Polymorphic Skill Induction** Where our method, PolySkill, improves is by integrating this process with a **polymorphic** skill structure. A critical preliminary step in our framework is that if the agent is operating on its first shopping website, it must first induce the high-level **abstract class**, `AbstractShoppingSite`, which provides a common ground of skills signature across shopping-related skills. Subsequently, during the skill induction phase, as the agent induces new skills on a specific site (e.g., `amazon.com`), it is guided to first register the corresponding function signature within the abstract class, and then define the concrete implementation within the site-specific class (e.g., `AmazonWebsite`, which inherits the `AbstractShoppingSite`). This reframes the induction prompt: rather than asking the LLM to directly induce skills, we instruct it to create a general abstract skill first and then implement the specific `search` method for an

---

**Algorithm 1** PolySkill: Polymorphic-Driven Skill Induction

---

1: **Input:** A sequence of tasks $\mathcal{Q} = \{q_1, \ldots, q_N\}$, LM Policy $\pi_{\mathcal{L}}$, LM Judge $V_{\mathcal{L}}$
2: **Initialize:** Dynamic skill library $\mathcal{K}_0 \leftarrow \emptyset$
3: **for** $t = 1, \ldots, N$ **do**
4:     Let $q_t$ be the current task from $\mathcal{Q}$.
5:     Define the agent's full action space: $\mathcal{A}_t \leftarrow \mathcal{A}_p \cup \mathcal{K}_{t-1}$.
6:     $\tau \leftarrow \text{ExecuteTask}(\pi_{\mathcal{L}}, q_t, \mathcal{A}_t)$             ▷ 1. Execute task to generate a trajectory
7:     **if** $V_{\mathcal{L}}(\tau, q_t) = 1$ **then**            ▷ 2. Verify trajectory correctness
8:         $\mathcal{K}_{new} \leftarrow \text{InduceSkill}(\pi_{\mathcal{L}}, \tau, \mathcal{K}_{t-1})$     ▷ 3. Induce new hierarchical skills
9:         $\mathcal{K}_t \leftarrow \mathcal{K}_{t-1} \cup \mathcal{K}_{new}$       ▷ Update skill library for the next task
10:     **else**
11:         $\mathcal{K}_t \leftarrow \mathcal{K}_{t-1}$       ▷ Skill library remains unchanged on failure
12:     **end if**
13: **end for**
14: **return** $\mathcal{K}_N$

---

`AmazonWebsite` class. This encourages the agent to learn skills that are not just locally effective but are structurally consistent implementations of a shared, domain-wide concept.

**Skill Learning on Unseen Websites** This polymorphic structure makes learning on new websites within a known category significantly more efficient. Imagine the agent has already formed the `AbstractShoppingSite` class derived from its initial interaction with `amazon.com`, and now visits `walmart.com` for the first time. It immediately recognizes Walmart as a shopping site and retrieves the abstract blueprint. This blueprint provides the agent with a clear set of exploration goals. Instead of randomly trying actions, it knows it needs to figure out how to concretely implement abstract skills like `search_product` and `add_to_cart` on this new site. Once the agent successfully searches for an item, it follows the standard induction process to create a new `WalmartWebsite` class, filling in the `search_product` method with the specific actions that worked. This guided approach accelerates learning by focusing the agent's efforts on mastering the essential skills defined in the abstract interface. The pseudocode is shown in Algorithm 1.

### 3.4 EVALUATION SETUP

We evaluate our induction process over baseline, in two different settings: **(1) Task-Defined Benchmarks** In standard benchmark settings, we apply this process within controlled environments, including **Mind2Web** (Deng et al., 2023) and **WebArena** (Zhou et al., 2024a). Here, the agent is presented with a predefined curriculum of tasks. Each successful trajectory provides the validated sequence of actions needed to implement a concrete skill method. This controlled approach allows us to rigorously measure how well the polymorphic representation facilitates generalization and transfer to unseen tasks, websites, or domains. **(2) Task-Free Continual Learning:** To assess the ultimate goal of agent autonomy, we also apply our framework in a task-free setting, similar to settings as Voyager (Wang et al., 2023) and SkillWeaver (Zheng et al., 2025). In this scenario, the agent explores websites on its own, **proposes its own goals**, and induces skills from its successful attempts. Crucially, we showed that our polymorphic hierarchy enables structured exploration (Murty et al., 2025; Gandhi & Neubig, 2025). The already-learned abstract domain classes act as a schema, providing a strong prior for what skills are worth discovering (Liang et al., 2025).

### 3.5 EVALUATION METRICS

We assess performance using five key metrics. Two are standard benchmarks adopted from prior work (Wang et al., 2024; 2025), while we introduce three new metrics to measure skills' utility. (1) **Task Success Rate (SR)** is the percentage of held-out tasks the agent completes successfully. This is the primary measure of overall performance. (2) **Number of Steps** is the average number of actions the agent takes to complete a task, where each primitive action and each call to a skill is counted as a single step. Fewer steps indicate higher efficiency. (3) **Skill Reusability** measures the number skill reused in new tasks. A high rate indicates the agent learns relevant, broadly applicable skills rather than overly niche ones. (4) **Task Coverage** measures the tasks that used at least one

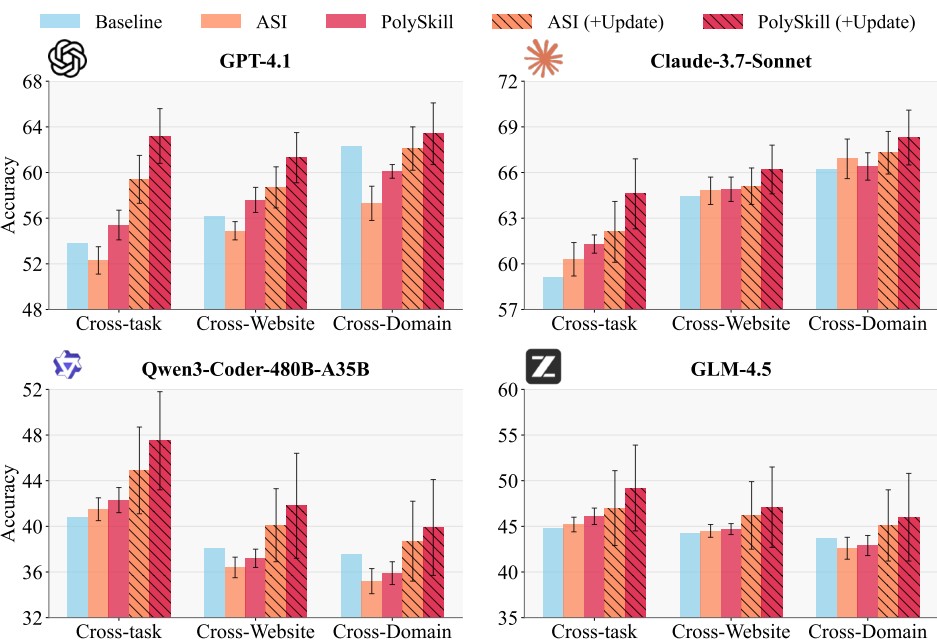

Figure 3: Performance comparison of PolySkill with baseline methods on the Mind2Web benchmark across four large language models. The y-axis shows task success rate (%). The three evaluation settings on the x-axis, Cross-task, Cross-Website, and Cross-Domain, represent increasing levels of generalization difficulty. Our method, *PolySkill*, consistently outperforms the ASI baseline, while the online continual update version, *PolySkill (+Update)*, achieves state-of-the-art performance across all models and settings. The performance gains are most significant in the challenging Cross-Domain scenario, highlighting our method's superior generalization. Error bars represent the standard error over three runs.

skill. This metric indicates whether the skills can be adaptive in actual test case scenarios. (5) **Skill Compositionality** measures how frequently the system reuses existing skills as building blocks for more complex tasks. A high score indicates an efficient and scalable learning process, as the system leverages its acquired knowledge rather than learning every new skill from scratch. Formal definitions for all metrics are provided in Appendix E.1.

## 4 EXPERIMENTS

In this section, we present a comprehensive evaluation of PolySkill. We first assess its performance and generalization capabilities in standard, task-defined benchmark settings. We then test its ability to learn autonomously in a more challenging task-free, exploratory scenario.

### 4.1 STANDARD BENCHMARK EVALUATION

**Benchmarks (1) Mind2Web** (Deng et al., 2023) To evaluate PolySkill on general web navigation scenarios We evaluate on Mind2Web, a comprehensive dataset spanning 137 websites across 31 domains. The dataset includes 2,350 tasks with annotated *cross-task*, *cross-website*, and *cross-domain* settings, providing diverse scenarios for skill learning and evaluation. We use the standard train-test split, holding out entire website categories for out-of-domain evaluation. Since the benchmark only comes with human trajectory data, we employs an automatic judge based on GPT-4.1 for measuring task success rate, which achieved an 85% agreement with human judgment (Xue et al., 2025). We employ the judge both in the skill induction stage and during the test time. **(2) WebArena** (Zhou et al., 2024a) It provides a realistic evaluation environment with fully functional websites across e-commerce, forums, development tools, and content management systems. The benchmark includes 812 tasks ranging from simple navigation to complex multi-step procedures, with automatic evaluation through functional correctness checks.

**Models and Baselines** For our experiments, we evaluate the performance on four foundation models: two closed-source (GPT-4.1 and Claude-3.7-Sonnet) and two open-source agentic models (Qwen3-Coder-480B-A35B (Qwen, 2025) and GLM-4.5 (GLM, 2025)). This selection encompasses leading proprietary models and open-source models that post-training specifically on agentic tasks, enabling a rigorous evaluation of agentic capabilities. We compare our method against three baselines: a standard agent with no skill induction (*Base*) and two leading skill induction frameworks, *ASI* (Wang et al., 2025) and *SkillWeaver* (Zheng et al., 2025). The ASI is induced online and applied to both Mind2Web and WebArena; however, for SkillWeaver, we only evaluate it on WebArena Subsets.

**Results.** For Mind2Web, as shown in Figure 3 (Results for WebArena are presented in Table 8). Our PolySkill outperforms ASI across both static and online settings. On GPT-4.1, PolySkill improves Cross-task accuracy from 52–53% (ASI) to 55–56%, and its online variant jumps to about 63%, compared to only 59% for ASI (+Update). In the hardest Cross-Domain setting, PolySkill (+Update) reaches 63–64% versus 62% for ASI (+Update). On Qwen3-Coder, PolySkill (+Update) improves Cross-task from 41.5% (ASI) to 47.5% and Cross-Domain from 35.2% to 39.9%. These consistent gains, particularly the significant jumps in Cross-Website, underscore PolySkill 's stronger generalization and the clear benefit of continual online updating over ASI.

## 4.2 ANALYSIS

To investigate how skills are induced during training, we use our newly proposed metrics to track the model's learning dynamics. This analysis builds on previous work that sought to understand these dynamics (Shah et al., 2025). For our study, we saved 20 model snapshots at regular intervals while training on the Mind2Web Cross-task and WebArena Shopping benchmarks. We then evaluated our metrics on each snapshot to observe how skills emerge over time.

**Relation between Skill Reusability and No. Steps** To validate the fundamental hypothesis that skill learning improves task efficiency, we analyze the relationship between Skill Reusability and the number of steps required for task completion. The results, presented in Figure 4, demonstrate a clear inverse correlation across all three methods: as skill reusability increases from 0% to over 20%, the average number of steps decreases substantially from approximately 6.1 to 3.3-4.4 steps. **PolySkill** achieves the highest Skill Reusability (reaching 20.4% by task 180), while maintaining competitive step reduction. Notably, ASI shows the most dramatic step reduction despite lower utilization rates, suggesting efficient skill application. This analysis confirms that learned skills directly translate to improved task efficiency, with higher utilization rates consistently leading to more streamlined task execu-

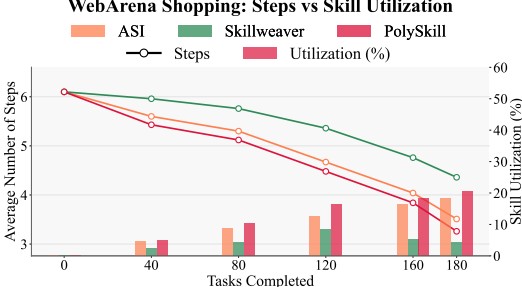

Figure 4: Relationship between skill reusability and task efficiency in WebArena shopping tasks. Lines show average steps (left y-axis) while bars show Skill Reusability (right y-axis) for ASI (orange), SkillWeaver (green), and PolySkill (red). Higher skill reusability correlates with fewer required steps, demonstrating improved task efficiency through learned skills.

tion. The strong correlation validates our core premise that skill induction and reuse are key drivers of agent performance improvement in complex web navigation tasks.

**Number of Skills Learned** We additional analyze the size of the induced skill library ("# Skill"). Ideally, an agent should identify a concise set of "polymorphic" skills that can be reused across various contexts, rather than memorizing a vast library of specific, non-transferable sub-routines. A lower skill count coupled with high accuracy indicates that the agent has successfully distilled the underlying logic of the tasks in certain layouts. We measure the number of skills by measuring how many unique skills are set aside, without considering the exact implementation per class. We include both the *w/o Update* and *w/ Update* settings.

As shown in Table 2, PolySkill demonstrates superior efficiency compared to ASI. In the static setting, PolySkill achieves higher accuracy (55.4% vs. 52.3% in Cross-task) while maintaining a smaller skill library (43 vs. 50 skills). This confirms

| Training Setting | | Evaluation Benchmark (SR % / Skill Usage %) | | | | | |
|---|---|---|---|---|---|---|---|
| Method | Iterations | WA Shopping | | AMZ | | Target | |
| Baseline | – | 37.4 | – | 47.3 | – | 60.5 | – |
| *1. Single-Domain Specialists* | | | | | | | |
| WA | 50 | 42.3 | 14.9 | 50.2 | 3.3 | 61.2 | 2.8 |
| AMZ | 50 | 38.1 | 2.7 | **69.5** | 48.3 | 61.5 | 3.0 |
| Target | 50 | 38.0 | 2.1 | 48.5 | 3.5 | 77.0 | 52.1 |
| *2. Sequential Curriculum* | | | | | | | |
| AMZ → WA | 75 + 75 | 40.2 | 12.3 | 65.3 | 42.7 | 62.5 | 3.1 |
| AMZ → Target → WA | 50 + 50 + 50 | 38.2 | 11.9 | 65.2 | 43.3 | **77.3** | 24.3 |
| Target → AMZ → WA | 50 + 50 + 50 | 39.5 | 11.5 | 66.1 | 40.8 | 69.2 | 18.9 |
| WA → Target → AMZ | 50 + 50 + 50 | 42.1 | 10.8 | 70.5 | 43.2 | 76.8 | 23.3 |
| SkillWeaver[*] | 150 | 39.8 | 8.6 | 64.4 | 25.2 | 74.2 | 18.3 |
| *3. Self-guided Exploration* | | | | | | | |
| AMZ + Target + WA | 150 | **43.1** | 14.6 | 66.7 | 36.4 | 75.2 | 19.4 |

Table 3: Performance in the task-free exploration setting for the Shopping Domain. [*] For the Skill-Weaver, we selected the best-performing curriculum in (WA → Target → AMZ).

our hypothesis that polymorphic skills are more robust to interface variations, allowing a single skill to cover scenarios that would require multiple distinct skills in ASI.

When online updates are enabled, the skill library naturally grows to accommodate novel environments; however, PolySkill + Update continues to outperform the baseline significantly. Notably, in the Cross-task setting, PolySkill + Update reaches state-of-the-art performance (63.2%) with only 47 skills, far fewer than the 66 skills required by ASI + Update. This suggests that PolySkill adapts by refining existing polymorphic definitions rather than blindly accumulating redundant skills.

| Method | Cross-task | | Cross-Website | | Cross-Domain | |
|---|---|---|---|---|---|---|
| | Acc ↑ | # Skill ↓ | Acc ↑ | # Skill ↓ | Acc ↑ | # Skill ↓ |
| Baseline | 53.8 | - | 56.2 | - | 62.3 | - |
| *Static Approaches* | | | | | | |
| ASI | 52.3 | 50 | 54.9 | 47 | 57.3 | 33 |
| **PolySkill** | 55.4 | 43 | 57.6 | 44 | 60.1 | 36 |
| *w/ Online Update* | | | | | | |
| ASI + Update | 59.4 | 66 | 58.7 | 71 | 62.1 | 66 |
| **PolySkill + Update** | **63.2** | **47** | **61.3** | **53** | **63.4** | **56** |

Table 2: Performance comparison on Mind2Web using GPT-4.1. We report success rate (Acc ↑) and # Skill ↓.

**Case Study: Continual Learning** To simulate how skill would be used and evolved in real-world scenarios, we investigate the agent's continual learning capabilities. The experiment begins with an agent whose skill library is initialized on the WebArena Shopping tasks. The agent then continues to perform online update via skill induction on new websites for Amazon and then Target, respectively (details in Appendix E.4). We performed the experiments 3 times to reduce the potential variance across run. In this setting, we are both interested in the positive transfer from the existing skill library in a similar domain. Also, we measure the WA Shopping performance after it has adapted to new website. This allows us to study potential catastrophic forgetting, where learning new knowledge can harm existing skills, a well-known challenge in continual learning (van de Ven et al., 2025).

The results, presented in Figure 5, highlight two critical advantages of PolySkill's polymorphic abstraction: First, when adapting to new domains like Amazon and Target, PolySkill effectively learns the required specialized skills, demonstrating strong positive transfer (orange and red curves). More importantly, it avoids the interference that reduces the ASI performance at the last. After specializing on the new sites, the ASI agent's performance on the original WebArena tasks degrades significantly. In contrast, PolySkill retains its general knowledge, with its performance on the original tasks remaining stable (blue curve). This results in a final +4.9% performance advantage over ASI on the benchmark, proving our method's ability to learn new skills without hurting existing ones.

## 4.3 FROM SPECIALIST TO EXPLORER: SKILL LEARNING IN AN EXPLORATIVE SETTING

To assess the goal of agent autonomy, we test our framework in an *explorative, continual learning* scenario. This setting extends beyond predefined tasks; instead, the agent explores websites independently, proposes its own goals, and acquires skills from its successful attempts (Zheng et al., 2025).

We designed an experiment to answer a central question: *Does an agent need a human-guided curriculum to learn general skills, or can its own exploration lead to versatile generalization?* To investigate this, we compared three learning paradigms using GPT-4.1 across shopping sites (`AMZ`, `Target`, `WA`) and developer platforms (`Github`, `GitLab`). First, we established two baselines to ground our comparison. **(1) Single-Domain Specialists,** the agent is trained on only one website, allowing us to measure the effects of over-specialization. **(2) Sequential Curriculum,** the agent follows a fixed, predefined order of websites, representing a standard pre-defined curriculum. And our proposed method, **(3) Self-guided Exploration**. This approach extends the self-proposing agent concept from SkillWeaver (Zheng et al., 2025) to a more challenging multi-website setting where the agent can freely choose which website to explore at each iteration. This autonomy is enabled by our core contribution, the polymorphic skill structure, which provides the necessary framework for the agent to autonomously structure, hone, and generalize its skills across these diverse platforms.

**Shopping Domains (OneStopShop, Amazon and Target)** As shown in Table 3, the choice of learning paradigm affects skill transfer. Single-domain specialists perform well on their home site (e.g., 77.0% SR on Target) but fail to transfer this knowledge, with Skill Reusability below 4% on other sites. The sequential curriculum improves transfer but is sensitive to the order of the curriculum. Critically, the fully autonomous agent achieves the highest general success rate (43.1%) on the held-out WA Shopping benchmark. One major finding from the results, showing WA OneStopShop actually transfer better to other websites, meaning it has much richer skills to be learned.

**Coding Platforms (Gitlab, Github)** We further test this hypothesis in a more challenging scenario, transferring skills between developer platforms. As detailed in Table 9, the self-guided agent once again demonstrates superior generalization. It achieves the highest success rate on the held-out GitLab benchmark (66.2%) while also attaining the best performance on GitHub (84.0%), proving its ability to master both domains concurrently.

This key finding across both experiments demonstrates that PolySkill's hierarchical abstraction enables an agent to autonomously build a robust and general skill set that outperforms methods relying on a handcrafted curriculum. It successfully learns to be a generalist explorer and skill refiner.

## 5 CONCLUSION

In this work, we introduced **PolySkill**, a framework that teaches web agents generalizable skills using the principle of *polymorphic abstraction*. By separating a skill's high-level intent from its website-specific implementation, our method enables agents to reuse capabilities across diverse environments. Our experiments show a major improvement in generalization, with **73%** of learned skills transferring to unseen websites, a improving improvement to the $<$**31%** achieved by prior methods. This approach resolves a key tension between the need for specialized skills and the adaptability required for the open web, and we confirmed its effectiveness on open-source agentic models, demonstrating its broad applicability.

Looking ahead, this framework opens several avenues for research, including handling highly dynamic websites, enabling skill sharing between agents, and incorporating human feedback. More broadly, the principle of polymorphism is not limited to web agents. It offers a powerful template for any agent that must operate in diverse environments sharing similar underlying structures (Xu et al., 2025)—from **robotics**, where skills must generalize across physical settings (Cheng et al., 2025; Liu et al., 2025a), to **tool use**, where software interfaces constantly evolve (Fei et al., 2025; Qiu et al., 2025). This work, therefore, provides a concrete step toward more robust and adaptive agents capable of *learning from experience* (Silver & Sutton, 2025) in diverse environments.

### 5.1 FUTURE WORK

To address the challenge of domains where task category boundaries are "fuzzy", a prime candidate for future work is moving from manual definition to autonomous skill clustering. Instead of relying on predefined labels, agents could propose abstract classes based on functional similarity, i.e. via soft-clustering of execution traces, allowing abstract interfaces to emerge organically. Furthermore, we envision enabling smaller models to acquire these polymorphic skills autonomously through RL . In this paradigm, the efficiency penalty ($\gamma$) would evolve from a fixed heuristic into a mechanism for reward shaping, allowing the agent to dynamically learn the optimal trade-off between exploration effort and skill reusability.

## 6 ACKNOWLEDGEMENT

We are grateful to our colleagues at Orby (now part of Uniphore) for their insightful discussions, comments, and support in setting up the evaluation environments during Simon's internship. We also thank fellow Orby interns Yijin Ni and Mengzhao Jia for their valuable discussions and feedback. Additionally, we appreciate the constructive feedback from the members of the CHATS lab on an early version of this manuscript. We thank Modal Lab and Thinking Machines Lab for their generous sponsorship of computing resources.

## REPRODUCIBILITY STATEMENT

The experiments on open-source models are run on an 8-H100 GPU cluster, and all calls for proprietary LLMs are made via their official API. All experiments are done via BrowserGym (de Chezelles et al., 2025) API, and experiment details are illustrated in Section 4 and Appendix E.

## ETHICS STATEMENT

**Real World Impacts** Smarter autonomous agents could be a big help in the real world. They have the potential to make computers much easier to use, especially for people with disabilities or those who lack technical skills. Agents could also automate many routine computer tasks, freeing people up for more creative work. While the agents in our paper aren't advanced enough for this yet, these future possibilities mean we need to think carefully about the social and economic impact on jobs.

Our own work here focuses on improving performance on research tests, so we don't believe it creates any immediate real-world harm. One clear concern, however, is that web agents might misuse websites or violate their terms of use and copyright. We take this seriously and will remove an agent's ability to access any site if requested by its owner.

**Bias and Safety** It's also very important to make sure these agents are fair and don't harm or exclude anyone. Before any agent is deployed, it needs to be checked carefully for hidden biases. Because agents can take actions in the world, they could cause more serious problems than a simple chatbot if proper safeguards aren't in place. More research is needed to understand and prevent these potential harms.

**Intended Use** The methods and models in this paper are intended for research purposes only. We use academic benchmarks like WEBARENA and MIND2WEB to measure progress in the field. The systems we've built are research prototypes and might not yet ready for real-world deployment, especially in high-stakes situations where errors could be costly.

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

Figure 5: A continual learning experiment showing PolySkill can prevent catastrophic forgetting. The experiment consists of two phases: initial skill library induced on the *in-domain* **WebArena Shopping** benchmark; followed by continual learning on *cross-webiste*: **Amazon** then **Target**. The orange and red lines (right y-axis) show that PolySkill learns the new websites more effectively than the ASI baseline. The blue lines (left y-axis) track performance on the original WA results throughout the experiment. Shaded regions represent the standard error across three runs.

This appendix provides comprehensive details supporting our main findings. Appendix A documents the use of large language models in our analysis. Appendix C discusses potential limitations of our work. Appendix E provides the detailed experiment details, including the definition of evaluation metrics, action space, and used dataset. Appendix F provides completed benchmark results across datasets and metrics.

## A LARGE LANGUAGE MODEL USAGE

We used large language models (LLMs) only for language refinement tasks, including grammar checking, phrasing adjustments, and enhancing readability. All scientific ideas, experiments, analyses, and results are the sole contributions of the authors. We only used LLMs for literature searching. It is important to note that the LLM was not involved in the ideation, research methodology, or experimental design. All research concepts, ideas, and analyses were developed and conducted by the authors.

## B EXTENDED RELATED WORKS

**Adaptive Web Agents** The central challenge for web agents is generalizing from training environments to the vast, unseen web. (Cheng et al., 2024; Zheng et al., 2024; Gou et al., 2025; Gu et al., 2025; Chen et al., 2025b). Although the Agashe et al. (2025) seems to have already solved most of the GUI agent tasks, the recent work Xue et al. (2025) still shows improvement needed to continue to improve agents in complex tasks and robustness. Similarly, Contextual Experience Replay discretizes trajectories into "experiences" (containing environment dynamics and skills) and retrieves relevant blocks to guide new episodes (Liu et al., 2025b).

**Skill Representation Formats** The representation of a "skill" is a central design choice in skill-learning agents. The simplest forms are low-level and lack abstraction. **Skills as Textual Descriptions** leverage LLMs to store procedural knowledge in natural language within a prompt library (Wang et al., 2024; Zhu et al., 2025). This offers flexibility but lacks the formal structure needed for

| ASI Skill Example | SkillWeaver Skill Example |
|---|---|
| ```python
def search_product(search_box_id: str, query: str):
    """
    Examples:
    search_product('595', 'sony bluetooth
headphones')
    """
    click(search_box_id)
    fill(search_box_id, query)
    keyboard_press('Enter')

def navigate_category(dropdown_id: str, category_id:
str):
    """
    Navigate to a product category by first clicking
a dropdown menu.
    Examples:
        navigate_category('5', '76')
    """
    click(dropdown_id)
    click(category_id)
``` | ```python
async def retrieve_categories(page):
    """
    Args:
    page : A Playwright `Page` instance that controls browser automation.

    Returns:
    list of dict
        A list of dictionaries where each dictionary contains the following
keys:
        - "name" (str): The name of the category.
        - "url" (str): The URL of the category.
    """
    await page.goto("/home-kitchen.html")
    category_elements = await page.query_selector_all("tabpanel menu menuitem")
    categories = []
    for element in category_elements:
        name = await (await element.get_property("text_content")).json_value()
        link_element = await element.query_selector("a")
        url = (
            await (await link_element.get_property("href")).json_value()
            if link_element
            else ""
        )
        categories.append({"name": name, "url": url})
    return categories
``` |

Table 4: Example of ASI Skills and SkillWeaver Skills.

reliable composition or verification. Similarly, **Skills as Action Traces** store successful action sequences or workflows from prior tasks (Wang et al., 2024; 2025). While easy to record, these traces implicitly encode the specifics of a single website's UI, making them fail when even minor elements change. A more structured approach represents **Skills as Programs**. This was pioneered in interactive environments like Minecraft, where agents build a library of reusable code snippets (Wang et al., 2023), and has been applied to web agents (Zheng et al., 2025) and software engineering (Chen et al., 2025a). However, these learned programs are typically *concrete implementations* tied to a single context, not inherently designed to handle variations across different websites fulfilling the same function.

**Architectures for Agentic Memory** Beyond the representation of individual skills, recent work has explored the broader architecture of agent memory to support long-horizon reasoning and learning. One line of work focuses on processing and structuring episodic experiences. For instance, Generative Agents (Park et al., 2023) structure memory as a stream of observations and use LLMs to distill high-level summaries to guide behavior. Other research focuses on memory management for efficiency and complex tasks. HiAgent organizes working memory hierarchically by subgoals, summarizing and replacing low-level traces to improve performance on long-horizon tasks (Hu et al., 2024). External-memory systems such as Mem0 (Chhikara et al., 2025) and MemP (Fang et al., 2025) augment agents with a persistent store managed by explicit add, update, and prune operations, yielding gains on dialogue and planning tasks.

## C  LIMITATIONS

We show that PolySkill also works for the latest agentic models for continual learning, which is a way to continuously improve these models with environment interaction settings. This also directly answers the claim that open-source models still fall behind in terms of skill induction (Shah et al., 2025). The promising next step would be to enable smaller agentic models to also utilize such skill acquisition behaviors, via further RL training on these memory environments.

## D  SKILL EXAMPLES

We show the ASI and SkillWeaver Skill Examples at Table 4; and the examples for PolySkill is at Table 1.

# E    EXPERIMENT DETAILS

## E.1    FORMAL DEFINITIONS OF EVALUATION METRICS

**Task Success Rate (SR)** This is the fraction of tasks in the evaluation set, $\mathcal{T}_{\text{test}}$, that the agent completed successfully. It is defined as:

$$\text{SR} = \frac{1}{|\mathcal{T}_{\text{test}}|} \sum_{T_j \in \mathcal{T}_{\text{test}}} \mathbb{I}(\text{task } T_j \text{ is successful})$$

where $\mathbb{I}(\cdot)$ is the indicator function which returns 1 if a task is successful and 0 otherwise.

**Number of Steps** This is the average number of actions the agent takes to complete a task, calculated exclusively over successful trajectories to measure efficiency. Let $\mathcal{D}_{\text{success}}$ be the set of trajectories for successfully completed tasks. For each trajectory $\tau \in \mathcal{D}_{\text{success}}$, let $|\tau|$ denote its length (number of actions). The average number of steps is:

$$\text{Number of Steps} = \frac{1}{|\mathcal{D}_{\text{success}}|} \sum_{\tau \in \mathcal{D}_{\text{success}}} |\tau|$$

**Skill Reusability** This metric measures the efficiency of the skill library itself by calculating the fraction of learned skills that were used at least once. Let $\mathcal{K}$ be the final library of learned skills and $\mathcal{D}_{\text{test}}$ be the set of all trajectories from the evaluation. The utilization is the fraction of skills $k \in \mathcal{K}$ that appear in at least one trajectory $\tau \in \mathcal{D}_{\text{test}}$:

$$\text{Skill Reusability} = \frac{|\{k \in \mathcal{K} \mid \exists \tau \in \mathcal{D}_{\text{test}}, k \in \tau\}|}{|\mathcal{K}|}$$

**Skill Adoption Rate** This metric measures the prevalence of skill-based behavior. Let $\mathcal{D}_{\text{test}}$ be the set of all test task trajectories and $\mathcal{K}$ be the library of induced skills. The adoption rate is the fraction of trajectories $\tau \in \mathcal{D}_{\text{test}}$ in which at least one skill $k \in \mathcal{K}$ was invoked:

$$\text{Skill Adoption Rate} = \frac{|\{\tau \in \mathcal{D}_{\text{test}} \mid \exists k \in \mathcal{K}, k \in \tau\}|}{|\mathcal{D}_{\text{test}}|}$$

**Skill Compositionality** This metric evaluates the hierarchical structure of the skill library. Let the final skill library be an ordered set $\mathcal{K} = \{k_1, \ldots, k_N\}$ of $N$ skills, where the index indicates creation time. For each skill $k_i$, let $\text{body}(k_i)$ be the set of non-primitive actions in its implementation. The compositionality is the average number of previously learned skills reused in each new skill:

$$\text{Skill Compositionality} = \frac{1}{N} \sum_{i=1}^{N} |\{k_j \in \text{body}(k_i) \mid j < i\}|$$

## E.2    ACTION SPACE

## E.3    DATASET

**Mind2Web** (Deng et al., 2023) is a large-scale, comprehensive benchmark designed for the development and evaluation of generalist web agents. It aims to measure an agent's ability to follow natural language instructions to perform complex tasks on any given website. The dataset is notable for its breadth, containing over 2,350 tasks spread across 137 different websites and spanning 31 distinct domains. The tasks are collected from real-world use cases and represent a wide array of common user activities, such as booking flights, managing online shopping carts, and configuring account settings. Each task instance consists of a high-level natural language instruction paired with a specific website. An agent's performance is evaluated based on its ability to successfully complete the instruction by generating a correct sequence of web-based actions, like clicking buttons, typing text, and selecting options from dropdowns. This benchmark's diversity in both tasks and domains makes it a robust tool for training and testing the generalization capabilities of autonomous web agents.

| Category | Action Type | Description |
|---|---|---|
| **Basic Actions** | `noop` | Do nothing |
| | `click(elem)` | Click at an element |
| | `hover(elem)` | Hover on an element |
| | `type(elem, text)` | Type to an element |
| | `press(key_comb)` | Press a key combination |
| | `scroll(dir)` | Scroll up and down |
| **Tab Operations** | `tab_focus(index)` | Focus on the i-th tab |
| | `new_tab` | Open a new tab |
| | `tab_close` | Close current tab |
| **Page Operations** | `go_back` | Visit the last URL |
| | `go_forward` | Undo go_back |
| | `goto(URL)` | Go to URL |

Table 5: The base primitive action space from BrowserGym (de Chezelles et al., 2025)

**WebArena** (Zhou et al., 2024a) is a realistic and reproducible web environment and benchmark designed to evaluate the functional capabilities of autonomous language agents in complex, goal-oriented scenarios. It features 811 distinct tasks that require agents to interact with five fully functional, self-hosted web applications: an e-commerce platform (OneStopShop), a social forum (Reddit), a collaborative software development platform (GitLab), a map service (OpenStreetMap), and a site administration dashboard for the e-commerce platform. Unlike benchmarks that focus on single-site interactions, many WebArena tasks are compositional, requiring the agent to navigate and synthesize information across multiple applications to achieve a final goal.

### E.4 REAL-WORLD WEBSITE TASKS

**Amazon (50 tasks)**

- Search for an "ergonomic office chair" from the brand 'Herman Miller' that costs between $500 and $1000.
- Find "27-inch 4k 144hz gaming monitors" that are eligible for Prime shipping.
- Look for an "air fryer toaster oven combo" with at least a 20-quart capacity and a customer rating of 4 stars or higher.
- Search for new "Sony WH-1000XM5 headphones" in the color 'Silver'.
- Find "silicone pet grooming gloves" designed specifically for cats with long hair.
- Search for 'Apple' "laptops" with at least 16GB of RAM and sort them by price from high to low.
- Find "robotic vacuums" from the brand 'iRobot' with a self-emptying feature, and sort the results by average customer review.
- Search for 'Vitamix' "blenders" with at least 1200 watts of power and sort them by price from low to high.
- Look for "science fiction books" by author 'Andy Weir' in paperback format and sort them by publication date, with newest first.
- Search for waterproof "men's winter jackets" in size 'Large' and sort by newest arrivals.
- Find "android tablets" with at least 64GB of storage that cost less than $200.
- Show me "men's running shoes" from the brand 'Brooks' in size 10.5.
- Find programmable "coffee makers" under $75 that have a 4-star rating or higher.
- Look for "QLED 4K televisions" from the brand 'Samsung' with a screen size of '65 inches'.

- Find waterproof "women's hiking boots" from the brand 'Merrell' in size 8 that are eligible for Prime shipping.
- Search for "Whey Isolate protein powder", filter by the flavor 'Unflavored', and a customer rating of 4 stars and up.
- Find "board games" suitable for "2 players" and for ages "12 and up" with an average customer review of at least 4 stars.
- Look for color-changing "smart home light bulbs" that are compatible with "Amazon Alexa".
- Find the cheapest "1TB NVMe SSD" with a minimum read speed of 5,000MB/s.
- From the brand 'Ninja', show me the least expensive "air fryer" with a minimum capacity of 6 quarts.
- Find the current price and the screen size in inches of the "Kindle Paperwhite (16 GB)".
- Go to the product page for the "Instant Pot Duo 7-in-1" and find both its capacity in quarts and its item weight.
- On the product page for the "Anker PowerCore 10000" power bank, find reviewers who mention the word "travel" and list their names along with the star rating they gave.
- For the "Sony WH-1000XM5" headphones, summarize the main criticisms mentioned in the 1-star and 2-star reviews.
- For the "Breville Barista Express Espresso Machine", find positive customer reviews (4-stars or higher) that specifically talk about the "steam wand".
- Summarize the positive comments from 5-star reviews for the "Kindle Scribe" that specifically mention the "writing experience".
- Summarize what customers in 3-star reviews say about the battery life of the "Apple AirPods Pro (2nd Generation)".
- Find the product page for the book "Dune" by Frank Herbert, list the available formats, and find the price of the Mass Market Paperback edition.
- Check the product page for the "Logitech MX Master 3S" mouse to see if it is compatible with both macOS and Windows 11.
- Find the "LEGO Star Wars Millennium Falcon" set (model 75257) and list its item dimensions and the manufacturer's recommended age range.
- Add two "Echo Dot (5th Gen)" devices in the color 'Charcoal' to my shopping cart.
- Find a "Hydro Flask 32 oz" Wide Mouth bottle in the color 'Olive' with a Flex Straw Cap and add it to the cart.
- Add two "Anker USB C to Lightning" cables (6ft) in 'White' to the shopping cart.
- Find the book "The Hobbit" by J.R.R. Tolkien and add both the paperback and the Kindle versions to your cart.
- Add one "The Lord of the Rings" paperback box set to the cart, view the cart, and then update the quantity to two.
- Add the "Breville Barista Express Espresso Machine" (model BES870XL) to my default wish list.
- Find the "Catan Seafarers" board game expansion and add it to your wish list.
- Add the "Sony WH-1000XM5" headphones in black to your wish list, then navigate to your wish list and sort it by "Price: High to Low".
- Create a new, private wish list named "Tech Gadgets".
- Add a "Samsung 980 Pro 2TB SSD" to the "Tech Gadgets" wish list with the comment "For new PC build".
- Navigate to the "Today's Deals" section and then filter to see only deals in the "Electronics" category.
- Go to the "Best Sellers in Books" page and then navigate to the "Science Fiction & Fantasy" sub-category.

- Show me the new releases in the "Electronics" category from the "Last 30 days".
- Go to the Amazon "Gift Cards" page and then find the "eGift Cards" section.
- Navigate to the customer service section and find help related to "A delivery, order or return".
- Find Amazon's return policy page and determine the return window for a new television.
- Find the help page on tracking packages and identify what to do if the tracking information is not updating.
- Navigate to the "Coupons" section of the website and filter to see coupons for "Grocery & Gourmet" products.
- Find "Amazon Basics" products in the "Home & Kitchen" category and sort the results by newest arrivals.
- Add a "Corsair K70 RGB PRO Mechanical Gaming Keyboard" with 'Cherry MX Red' switches to the cart and then proceed to the checkout page.

**Target** (50 tasks)

- Search for an "ergonomic office chair" from the brand 'Threshold' that costs between $100 and $250.
- Find "27-inch 4K computer monitors" that are available for same-day delivery.
- Look for an "air fryer toaster oven combo" with at least a 20-quart capacity and a guest rating of 4 stars or higher.
- Search for new "Beats Studio Pro headphones" in the color 'Navy'.
- Find "kitchen towels" from the brand 'Hearth Hand with Magnolia' that are 100% cotton.
- Search for "laptops" with at least 16GB of RAM from the brand 'HP' and sort them by price from high to low.
- Find "robotic vacuums" from the brand 'iRobot' with a self-emptying feature, and sort the results by "Best sellers".
- Search for "single-serve coffee makers" from the brand 'Keurig' and sort them by price from low to high.
- Look for "LEGO Creator 3-in-1" sets and sort them by "Newest".
- Search for "men's winter jackets" that are waterproof and in size 'Large', then sort by guest rating.
- Find "patio conversation sets" that include a fire pit and cost less than $750.
- Show me "women's athletic shorts" from the brand 'All in Motion' in a size 'Medium'.
- Find "electric toothbrushes" from the brand 'Philips Sonicare' with a guest rating of 4 stars or higher.
- Look for "OLED 4K smart TVs" from the brand 'Sony' with a screen size of '65 inches', and filter for "Order Pickup".
- Find "men's hiking boots" from the brand 'Merrell' in size 10 that are available for "Same Day Delivery".
- Search for "blenders", filter for the 'Ninja' brand, models with Auto-iQ, and a guest rating of 5 stars.
- Find "nursery gliders" from the brand 'DaVinci' that are currently on sale and available in the color 'Gray'.
- Look for "Nintendo Switch" games that are rated "Everyone 10+" and have the "Action" genre selected.
- Find the cheapest "8-cube organizer shelf" from the brand 'Brightroom' available in 'White'.
- Show me the least expensive "carry-on luggage" that has a hardside exterior, spinner wheels, and is from the brand 'Made By Design'.

- Find the current price and overall dimensions of the "Threshold designed with Studio McGee Herriman Wooden Console Table".
- Go to the product page for the "Keurig K-Mini Single-Serve Coffee Maker" and find out if it has an auto shut-off feature and its water tank capacity from the item details.
- On the product page for "Good  Gather Organic Blue Corn Tortilla Chips", check the nutrition details to see the amount of sodium and dietary fiber per serving.
- For the "Beats Studio Pro" headphones, go to the Q&A section and find out the reported battery life and if they come with a carrying case.
- Find customer reviews for the "Dyson V8 Origin Cordless Stick Vacuum" that mention its performance on both "pet hair" and "hardwood floors".
- Summarize the positive comments from 5-star reviews for the "Ninja Foodi 6-in-1 8qt 2-Basket Air Fryer" that talk about "ease of cleaning" and "cooking speed".
- What do customers in the 1- and 2-star reviews say about the fit and fabric quality of the "A New Day Women's High-Rise Slim Fit Ankle Jeans"?
- Find the product page for the book "Fourth Wing" by Rebecca Yarros. What is the listed genre and the total number of pages in the item details?
- Check the product page for the "Apple iPad 10.9-inch (10th Generation)" to see how many colors it is available in and if it is compatible with the 1st generation Apple Pencil.
- Find the "Fisher-Price Laugh  Learn Smart Stages Piggy Bank" toy and identify the manufacturer's suggested age range and the number of batteries required from the product details.
- Add two cartons of "Good  Gather Grade A Large Eggs - 12ct" to my shopping cart.
- Find a "Stanley 40oz Stainless Steel H2.0 FlowState Quencher Tumbler" in the color 'Fog' and add it to the cart, then change the quantity to 2.
- Add two "up  up 50-load lavender-scented laundry detergent" containers and one "downy fabric softener" to the shopping cart.
- Find a "Hearth  Hand with Magnolia" brand scented candle and add two different scents to the cart.
- Add one bag of "Good  Gather Organic Baby Carrots" and one "Good  Gather Classic Hummus" to the cart, then proceed to checkout but do not place the order.
- Create a new baby registry for an expected arrival date of June 1, 2026, and set it to be private.
- After creating a baby registry, add both the "Graco 4Ever DLX 4-in-1 Convertible Car Seat" and the "Owlet Dream Sock Baby Monitor" to it.
- Create a new custom list named "College Dorm Essentials".
- Add a "Room Essentials Twin/Twin XL Microfiber Comforter" in gray to your "College Dorm Essentials" list.
- Add a "Brightroom 3 Tier Metal Utility Cart" in white to your "College Dorm Essentials" list with a note that says "For bathroom and shower supplies".
- Navigate to the "Weekly Ad" section and find a deal on "Good  Gather" brand ground beef.
- Go to the "Clearance" section of the website and filter for "Home Deals" that are discounted by 50% or more.
- Find the "Target Circle" offers page and clip a deal for "20
- Go to the "Gift Cards" page and find the section for "Thank You" themed e-gift cards.
- Find the store locator page and check the guest service hours and Starbucks hours for the Target store in Somerville, MA.
- Find information on Target's return policy for electronics that have been opened and used.
- Navigate to the "RedCard" page and find the listed benefits of having a Reloadable RedCard versus a Debit or Credit RedCard.
- From the homepage, navigate to the "Toys" category and filter for "Action Figures  Playsets" from the brand "Marvel".

- Find the "Top Deals" in the "Home" category and sort them by "Discount: High-low".
- Add a "Threshold 16pc Stoneware Avesta Dinnerware Set" to your cart, then proceed to checkout and select "Store Pickup" for the Medford, MA location.

**Github (75 tasks)** Note that for the GitHub tasks, we force the tasks to be navigational or tasks that won't codeify existing code. We created dummy account for performing the experiments.

- Go to your profile and change your bio to "Building the future, one line of code at a time".
- Set your personal website URL in your profile to "https://www.github.com".
- Set your current status to "Focusing on a project" with the busy indicator on.
- Find the "microsoft/vscode" repository and star it.
- Create a new, public repository named "learning-python".
- Create a new, private repository named "project-secrets" and initialize it with a README file.
- Create a new public repository named "website-template" from the "jekyll/jekyll-now" template.
- Fork the "facebook/react" repository to your own account.
- In your "learning-python" repository, create a new file named "hello.py" with the content "print('Hello, World!')".
- Edit the README.md file in your "learning-python" repository to add the description "My personal repository for learning Python".
- In your "learning-python" repository, change the LICENSE file to the "MIT License".
- Find the SSH URL to clone the "tensorflow/tensorflow" repository.
- Navigate to the "twbs/bootstrap" repository and view its commit history for the "main" branch.
- In the "torvalds/linux" repository, find out how many commits were made by "Linus Torvalds" in the last month.
- Find who the top contributor is for the "openai/gpt-3" repository.
- List the names and number of commits for the top 3 contributors to the "google/gvisor" repository.
- Go to the "Explore" page to see trending repositories.
- Search for repositories with the topic "web-agent" written in the "Python" language.
- Find and navigate to your notifications page.
- Go to the page that lists all pull requests that are assigned to you.
- Navigate to the page that lists all issues where your review is requested.
- Find the "kubernetes/kubernetes" repository and view all its open issues.
- In the "microsoft/terminal" repository, filter the issues to show only those with the "bug" label.
- Create a new issue in your "learning-python" repository with the title "Need to add a requirements.txt file".
- In your "learning-python" repository, create a new issue titled "Refactor hello.py" and assign it to yourself.
- Create an issue in your "learning-python" repository with the title "Add data structures examples", and add the labels "enhancement" and "help wanted".
- In the "atom/atom" repository, find the oldest open issue and leave the comment "Is this still relevant?".
- Find a pull request in the "expressjs/express" repository that updates dependencies and post the comment "LGTM!" on it.

- In your "website-template" repository, create a new branch named "feature/add-contact-page".

- Create a pull request in your "website-template" repository to merge the "feature/add-contact-page" branch into the "main" branch.

- Create a pull request in your "website-template" repository with the title "Update Homepage", merging from "develop" to "main", and assign yourself as the reviewer.

- In your "learning-python" repository, add "torvalds" as a collaborator with "Read" access.

- Create a new milestone in your "learning-python" repository titled "Version 1.0 Release".

- Set the due date for the "Version 1.0 Release" milestone in your "learning-python" repository to be one month from today.

- Create an issue titled "Finalize documentation" in your "learning-python" repository and assign it to the "Version 1.0 Release" milestone.

- Find all closed issues in the "docker/compose" repository that mention "network".

- View your starred repositories and sort them by "Recently starred".

- Create a new organization named "My-Awesome-Startup-Org".

- Invite the user "octocat" to be a member of your "My-Awesome-Startup-Org" organization.

- Find the "sveltejs/svelte" repository and navigate to its "Projects" board.

- In the "NationalSecurityAgency/ghidra" repository, find the total number of watchers.

- Change the description of your "learning-python" repository to "A repository to track my Python learning journey and projects".

- Enable GitHub Pages for your "website-template" repository on the "main" branch.

- In the "numpy/numpy" repository, find the pull request with the most comments.

- In your "learning-python" repository, protect the "main" branch to require a pull request review before merging.

- Find the "actions/checkout" repository and view its different tags/releases.

- Create a new private repository and import the "git/git" repository into it.

- Follow the user "Linus Torvalds" (torvalds) on GitHub.

- In the "rust-lang/rust" repository, find all issues with the "A-async-await" label that are currently open.

- Create a pull request in your "learning-python" repository from a new branch called "fix/typo-in-readme" to "main" that corrects a spelling mistake in the README.

### E.5    PROMPTS

**Skill Induction Prompt**    We follow similar prompts as the ASI paper (Wang et al., 2025).

---

**Prompt for Skill Induction**

You are a proficient software engineer. Your task is to (1) summarize reusable functions as APIs from the provided action trajectories, and (2) rewrite the trajecoties using the reusable functions you generated in (1).

```
Tasks: {task}

Domains: {domain_url}

Trajectories: {
    Planner Step 1
    Executor Step 1

    Planner Step 2
    Executor Step 2

    Planner Step 3
    Executor Step 3
    ...
}
```

For (1), from the provided examples about the same task, you job is to generate Python functions that can be reused to solve (part of) these tasks. The functions should have mediocre complexity: (i) containing at least three actions and not too simple (e.g., a single line of code), (ii) not too complex (e.g., more than 10 lines of code), and should be general enough to be applied to other similar tasks. The arguments to these functions should be common variables (such as strings and lists), avoid using complex inputs such as another function. Please include 'Args', 'Returns', and 'Examples' in the function documentation.

For (2), write the instruction and rewritten code of each example. Do not include the answer response or example-specific information in the rewritten code. Pay attention to whether all link IDs are available before specifying them in the generated functions. If you use 'send_msg_to_user', make sure the message is decided within the function, instead of provided as an argument.

Make sure each function contains no less than 2 steps, and no more than 5 steps; to keep the functions simple and task-oriented. You can generate zero, one, or multiple functions depending on the provided examples.

---

**Judge Prompt** For Judge, we follow the prompt used in Online-Mind2Web (Xue et al., 2025), which showed over 85% agreement with human annotators.

## WebJudge - Key Point Identification

You are an expert tasked with analyzing a given task to identify the key points explicitly stated in the task description.

**Objective**: Carefully analyze the task description and extract the critical elements explicitly mentioned in the task for achieving its goal.

**Instructions**:
1. Read the task description carefully.
2. Identify and extract **key points** directly stated in the task description.
- A **key point** is a critical element, condition, or step explicitly mentioned in the task description.
- Do not infer or add any unstated elements.
- Words such as "best," "highest," "cheapest," "latest," "most recent," "lowest," "closest," "highest-rated," "largest," and "newest" must go through the sort function (e.g., the key point should be "Filter by highest").
**Respond with**:
- **Key Points**: A numbered list of the explicit key points for completing this task, one per line, without explanations or additional details.

Task: {task}

## WebJudge - Key Screenshot Identification

You are an expert evaluator tasked with determining whether an image contains information about the necessary steps to complete a task.

**Objective**: Analyze the provided image and decide if it shows essential steps or evidence required for completing the task. Use your reasoning to explain your decision before assigning a score.

**Instructions**:
1. Provide a detailed description of the image, including its contents, visible elements, text (if any), and any notable features.
2. Carefully examine the image and evaluate whether it contains necessary steps or evidence crucial to task completion:
- Identify key points that could be relevant to task completion, such as actions, progress indicators, tool usage, applied filters, or step-by-step instructions.
- Does the image show actions, progress indicators, or critical information directly related to completing the task?
- Is this information indispensable for understanding or ensuring task success?
- If the image contains partial but relevant information, consider its usefulness rather than dismissing it outright.

3. Provide your response in the following format:
### Reasoning: [Your explanation]
### Score: [1-5]

**Task**: {task}

**Key Points for Task Completion**: {key points}

The snapshot of the web page is shown in the image.

---

**WebJudge - Outcome Judgement**

You are an expert in evaluating the performance of a web navigation agent. The agent is designed to help a human user navigate a website to complete a task. Given the user's task, the agent's action history, key points for task completion, some potentially important web pages in the agent's trajectory and their reasons, your goal is to determine whether the agent has completed the task and achieved all requirements.

Your response must strictly follow the following evaluation criteria!

**\*Important Evaluation Criteria\*:**
1: The filtered results must be displayed correctly. If filters were not properly applied (i.e., missing selection, missing confirmation, or no visible effect in results), it should be considered a failure.
2: You must carefully check whether these snapshots and action history meet these key points. Ensure that specific filter conditions, such as "best," "highest," "cheapest," "latest," "most recent," "lowest," "closest," "highest-rated," "largest," and "newest" are correctly applied using the filter function (e.g., sort function).
3: Certain key points or requirements should be applied by the filter. Otherwise, a search with all requirements as input will be deemed a failure since it cannot guarantee that all results meet the requirements!
4: If the task requires filtering by a specific range of money, years, or the number of beds and bathrooms, the applied filter must exactly match the given requirement. Any deviation results in failure. To ensure the task is successful, the applied filter must precisely match the specified range without being too broad or too narrow.
5: Some tasks require a submission action or a display of results to be considered successful. Repeat actions or actions that do not lead to a visible result should be considered a failure.
6: If the agent loops through a sequence of actions that do not make progress toward the goal (including failing to click "Save" or "Submit," etc.), it should be considered a failure.

Format your response into two lines as shown below:
**Thoughts:** ¡your thoughts and reasoning process based on double-checking each key points and the evaluation criteria¿
**Status:** "success" or "failure"

User Task: {task}

Key Points: {key points}

Action History: {action history}

The potentially important snapshots of the webpage in the agent's trajectory and their reasons: {thoughts}

---

# F COMPLETE RESULTS

We put our complete results at Table 6 and Table 8.

Table 6: Performance comparison on Mind2Web benchmark. Results show success rates (%) across different generalization scenarios. Green indicates improvement and red indicates degradation from baseline. Best results in **bold**.

| Method | GPT-4.1 (Training: 1009 tasks) | | | Claude-3.7-Sonnet (Training: 1009 tasks) | | |
|---|---|---|---|---|---|---|
| | Cross-task (252) | Cross-Website (177) | Cross-Domain (912) | Cross-task | Cross-Website | Cross-Domain |
| Baseline | 53.8 | 56.2 | 62.3 | 59.1 | 64.4 | 66.2 |
| ASI (All skills) | $52.3_{\pm1.2}$ (-1.5) | $54.9_{\pm0.8}$ (-1.3) | $57.3_{\pm1.5}$ (-2.9) | $60.3_{\pm1.1}$ (+1.2) | $64.8_{\pm0.9}$ (+0.4) | $66.9_{\pm1.3}$ (+0.7) |
| ASI (Same domain) | $55.2_{\pm1.4}$ (+1.4) | $57.0_{\pm1.0}$ (+0.8) | N/A | $60.9_{\pm1.6}$ (+1.8) | $64.8_{\pm0.7}$ (+0.4) | N/A |
| ASI (Same sub-domain) | $56.3_{\pm1.7}$ (+2.5) | N/A | N/A | $61.2_{\pm1.3}$ (+2.1) | N/A | N/A |
| ASI (+Update) | $\underline{59.4}_{\pm2.1}$ (+5.6) | $58.7_{\pm1.8}$ (+2.5) | $62.1_{\pm1.9}$ (+1.9) | $62.1_{\pm2.0}$ (+3.0) | $65.1_{\pm1.2}$ (+0.7) | $67.3_{\pm1.4}$ (+1.1) |
| **PolySkill (All Skills)** | $55.4_{\pm1.3}$ (+1.6) | $57.6_{\pm1.1}$ (+1.4) | $60.1_{\pm0.6}$ (-0.1) | $61.3_{\pm0.6}$ (+2.2) | $64.9_{\pm0.8}$ (+0.5) | $66.4_{\pm0.9}$ (+0.2) |
| **PolySkill (Same domain)** | $58.3_{\pm1.8}$ (+4.5) | $58.9_{\pm1.5}$ (+2.7) | N/A | $62.0_{\pm1.7}$ (+2.9) | $64.9_{\pm0.8}$ (+0.5) | N/A |
| **PolySkill (Same sub-dom.)** | $58.6_{\pm2.0}$ (+4.8) | N/A | N/A | $62.3_{\pm1.9}$ (+3.2) | N/A | N/A |
| **PolySkill (+Update)** | $\mathbf{63.2}_{\pm2.4}$ (+9.4) | $\mathbf{61.3}_{\pm2.2}$ (+5.1) | $\mathbf{63.4}_{\pm2.7}$ (+3.2) | $\mathbf{64.6}_{\pm2.3}$ (+5.5) | $\mathbf{66.2}_{\pm1.6}$ (+1.8) | $\mathbf{68.3}_{\pm1.8}$ (+2.1) |

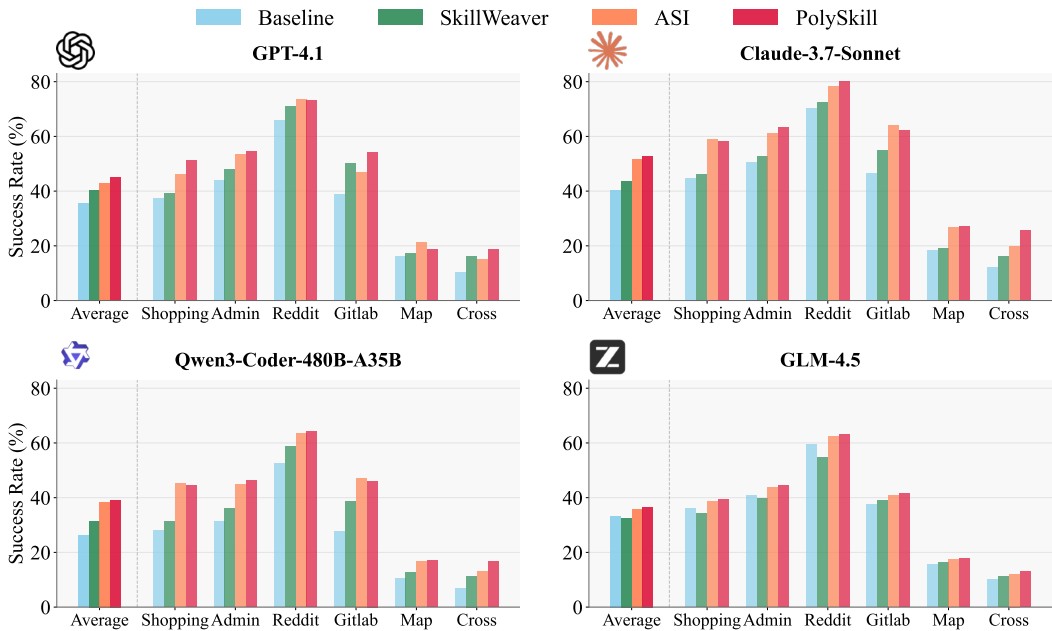

Figure 6: Overall performance comparison of PolySkill with baselines on the WebArena benchmark across four leading language models. The x-axis shows different website categories, with the leftmost **Average** group representing the primary overall result. PolySkill consistently achieves the highest average success rate on all models, demonstrating its effectiveness on complex, realistic web tasks. Notably, our method surpasses strong skill-learning baselines like ASI and SkillWeaver, with the most significant gains observed on the powerful GPT-4.1 and Claude-3.7-Sonnet models.

Table 7: Mind2Web results for open-source models. These represent the first evaluation of skill learning methods on open-source agentic models.

| Method | Qwen3-Coder-480B-A35B (Training: 1009 tasks) | | | GLM-4.5 (Training: 1009 tasks) | | |
|---|---|---|---|---|---|---|
| | Cross-task (252) | Cross-Website (177) | Cross-Domain (912) | Cross-task | Cross-Website | Cross-Domain |
| Baseline | 40.8 | 38.1 | 37.5 | 44.8 | 44.2 | 43.7 |
| ASI | $41.5_{\pm1.0}$ (+0.7) | $36.4_{\pm0.9}$ (-1.7) | $35.2_{\pm1.1}$ (-2.3) | $45.2_{\pm0.8}$ (+0.4) | $44.5_{\pm0.7}$ (+0.3) | $42.6_{\pm1.2}$ (-1.1) |
| ASI (+Update) | $44.9_{\pm3.8}$ (+4.1) | $40.1_{\pm3.2}$ (+2.0) | $38.7_{\pm3.5}$ (+1.2) | $47.0_{\pm4.1}$ (+2.2) | $46.2_{\pm3.7}$ (+2.0) | $45.1_{\pm3.9}$ (+1.4) |
| **PolySkill** | $42.3_{\pm1.1}$ (+1.5) | $37.2_{\pm0.8}$ (-0.9) | $35.9_{\pm1.0}$ (-1.6) | $46.1_{\pm0.9}$ (+1.3) | $44.7_{\pm0.6}$ (+0.5) | $42.9_{\pm1.1}$ (-0.8) |
| **PolySkill (+Update)** | $\mathbf{47.5}_{\pm4.3}$ (+6.7) | $\mathbf{41.8}_{\pm4.6}$ (+3.7) | $\mathbf{39.9}_{\pm4.2}$ (+2.4) | $\mathbf{49.2}_{\pm4.7}$ (+4.4) | $\mathbf{47.1}_{\pm4.4}$ (+2.9) | $\mathbf{46.0}_{\pm4.8}$ (+2.3) |

## F.1 COST AND EFFICIENCY ANALYSIS

To evaluate the economic feasibility and efficiency of our framework, we conducted a detailed cost analysis measuring token consumption. We calculated the average token usage (input + output) per instance in the Mind2Web cross-task setting using GPT-4.1. The breakdown distinguishes between the *Action Generation* phase (inference) and the overhead costs associated with *Verification* and *Skill Induction*.

As shown in Table 10, PolySkill introduces an overhead for skill induction and verification, resulting in a higher total token count for the initial run compared to the no-skill baseline. However, this investment yields significant efficiency gains:

1. **Reduced Inference Cost:** PolySkill reduces token consumption during the Action Generation phase by approximately **10.2%** compared to the Baseline (from ∼55.4k to ∼49.7k tokens) and **7.7%** compared to ASI. This is attributed to the polymorphic skills allowing the agent to solve tasks in fewer steps with higher precision.

2. **Amortized Efficiency:** Crucially, the costs for *Verification* and *Skill Induction* are primarily **one-time (or amortized) costs**. Once a concrete implementation for a website is induced, it is stored in the library and reused for all future tasks on that site. Conse-

Table 8: Performance comparison on WebArena benchmark. Results show success rates (%) across different website categories. Best results in **bold**, second best underlined. Experiments with − are pending completion.

| Method
# Instances | Shopping
187 | Admin
182 | Reddit
106 | Gitlab
180 | Map
109 | Cross
48 | Average
812 |
|---|---|---|---|---|---|---|---|
| *GPT-4.1* | | | | | | | |
| Baseline | 37.4 | 44.0 | 66.0 | 38.9 | 16.4 | 10.3 | 38.5 |
| SkillWeaver | 39.3 | 48.2 | 71.2 | 50.3 | 17.2 | 16.3 | 43.6 |
| ASI | 46.3 | 53.6 | 73.7 | 46.8 | 21.5 | 15.1 | 46.5 |
| **PolySkill (Ours)** | **51.4** | **54.8** | 73.2 | **54.2** | 18.9 | **18.9** | **49.3** |
| Δ vs ASI | +5.1 | +1.2 | -0.5 | +7.4 | -2.6 | +3.8 | +2.8 |
| *Claude-3.7-Sonnet* | | | | | | | |
| Baseline | 44.7 | 50.8 | 70.2 | 46.7 | 18.3 | 12.1 | 45.6 |
| SkillWeaver | 46.2 | 52.7 | 72.5 | 55.1 | 19.1 | 16.2 | 47.5 |
| ASI | **59.1** | 61.3 | 78.5 | **64.2** | 26.7 | 20.1 | 55.8 |
| **PolySkill (Ours)** | 58.3 | **63.5** | **80.4** | 62.5 | **27.4** | **25.6** | **59.5** |
| Δ vs ASI | -0.8 | +2.2 | +1.9 | -1.7 | +0.7 | +5.5 | +3.7 |
| *Qwen3-Coder-480B-A35B-Instruct* | | | | | | | |
| Baseline | 28.1 | 31.2 | 52.4 | 27.7 | 10.5 | 6.8 | 34.4 |
| SkillWeaver | 31.3 | 36.1 | 58.7 | 38.5 | 12.7 | 11.4 | 38.1 |
| ASI | **45.1** | 44.8 | 63.5 | **47.2** | 16.7 | 13.1 | 43.9 |
| **PolySkill (Ours)** | 44.4 | **46.3** | **64.2** | 46.1 | **17.1** | **16.8** | **45.2** |
| Δ vs ASI | -0.7 | +1.5 | +0.7 | -1.1 | +0.4 | +3.7 | +1.3 |
| *GLM-4.5* | | | | | | | |
| Baseline | 36.1 | 40.9 | 59.4 | 37.5 | 15.7 | 10.0 | 36.2 |
| SkillWeaver | 34.2 | 39.7 | 54.8 | 39.1 | 16.4 | 11.2 | 33.6 |
| ASI | 38.6 | 43.7 | 62.5 | 40.8 | 17.3 | 12.1 | 38.9 |
| **PolySkill (Ours)** | **39.4** | **44.5** | **63.2** | **41.6** | **17.8** | **13.0** | **39.8** |
| Δ vs ASI | +0.8 | +0.8 | +0.7 | +0.8 | +0.5 | +0.9 | +0.9 |

quently, as the number of tasks performed on a website increases, the average cost per task converges toward the significantly lower Action Generation cost, making PolySkill highly cost-effective for long-term deployment.

# G    EXTENDED METHODOLOGY

| Training Setting | | GitLab | | Github | |
| --- | --- | --- | --- | --- | --- |
| Method | Iters | SR % | Skill Usage % | SR % | Skill Usage % |
| Baseline | – | 38.9 | – | 66.7 | – |
| *1. Single-Domain Specialists* | | | | | |
| GitLab | 50 | 65.5 | 59.2 | 71.5 | 12.8 |
| Github | 50 | 40.2 | 3.8 | 81.5 | 54.1 |
| *2. Sequential Curriculum* | | | | | |
| Gitlab → Github | 50 + 50 | 48.3 | 20.5 | 80.1 | 51.9 |
| Github → GitLab | 50 + 50 | 62.1 | 45.3 | 77.8 | 48.6 |
| *3. Self-guided Exploration* | | | | | |
| Github + GitLab | 100 | **66.2** | 48.1 | **84.0** | 39.5 |

Table 9: Skill transfer results between software development platforms. This experiment highlights the challenge of transferring skills from a higher-performing domain (GitHub) to a more complex one (GitLab). Self-guided exploration, which learns from both domains concurrently, achieves the highest success rate on the held-out GitLab benchmark.

Table 10: Average token consumption per task instance on Mind2Web. PolySkill significantly reduces token usage during action generation, leading to lower long-term costs despite the initial overhead of skill induction.

| Method | Action Gen. | Verification | Skill Induction | Total Tokens |
| --- | --- | --- | --- | --- |
| Baseline (No Skills) | 55,367.4 | – | – | 55,367.4 |
| ASI | 53,840.5 | 4,673.8 | 5,773.2 | 64,287.5 |
| **PolySkill** | **49,710.3** | 4,963.2 | 8,476.4 | 63,149.9 |

**Algorithm 2** PolySkill in a Task-Free Setting (illustrating changes from Algorithm 1)

1:    – **Input:** A sequence of tasks $\mathcal{Q} = \{q_1, \ldots, q_N\}$, LM Policy $\pi_\mathcal{L}$, LM Judge $V_\mathcal{L}$
2:    + **Input:** Number of exploration steps $T$, LM Policy $\pi_\mathcal{L}$, Auto Judge $V_\mathcal{L}$
3:    – **Initialize:** Dynamic skill library $\mathcal{K}_0 \leftarrow \emptyset$
4:    + **Initialize:** Dynamic skill library $\mathcal{K}_0 \leftarrow \emptyset$, Initial observation $o_0$
5:    –**for** $t = 1, \ldots, N$ **do**

6:    +**for** $t = 1, \ldots, T$ **do**

9:    –    Let $q_t$ be the current task from $\mathcal{Q}$.
10:    +    $q_{proposed} \leftarrow \text{ProposeTask}(\pi_\mathcal{L}, o_{t-1}, \mathcal{K}_{t-1})$
11:    Define the agent's full action space: $\mathcal{A}_t \leftarrow \mathcal{A}_p \cup \mathcal{K}_{t-1}$.
12:    –    $\tau \leftarrow \text{ExecuteTask}(\pi_\mathcal{L}, q_t, \mathcal{A}_t)$
13:    +    $\tau \leftarrow \text{ExecuteTask}(\pi_\mathcal{L}, q_{proposed}, \mathcal{A}_t)$
14:    – **if** $V_\mathcal{L}(\tau, q_t) = 1$ **then**

16:    + **if** $V_\mathcal{L}(\tau, q_{proposed}) = 1$ **then**

18:    $k_{new} \leftarrow \text{InduceSkill}(\pi_\mathcal{L}, \tau, \mathcal{K}_{t-1})$
19:    $\mathcal{K}_t \leftarrow \mathcal{K}_{t-1} \cup \{k_{new}\}$
20:
21:    $\mathcal{K}_t \leftarrow \mathcal{K}_{t-1}$
22:
23:    +    $o_t \leftarrow \text{GetLastObservation}(\tau)$
24:
25:

