# OpenReview forum: "PolySkill: Learning Generalizable Skills Through Polymorphic Abstraction For Continual Learning"
_ICLR.cc/2026/Conference — ICLR 2026 Poster_

### Official Review · Reviewer_M5TL · 2025-10-25

**Soundness:** 3
**Presentation:** 3
**Contribution:** 3
**Rating:** 6
**Confidence:** 4

**Summary:**

This paper proposes a method to induce skills that transfer across websites, by decoupling their abstract goals and concrete implementations. Both experiments on static benchmarks and free-form continual learning settings show the empirical benefits of the proposed method.

**Strengths:**

**1. Importance of Skill Transferrability.**
> This paper tackles a critical limitation of current agent skill learning studies – transferability across websites.

**2. Empirical Improvements throughout Experiments.**
> The paper evaluates on both (i) static benchmarks and (ii) open-ended exploration settings. The proposed method shows improvements in success rate, efficiency, and skill reusability for (i); as well as greater task coverage and skill compositionality for (ii).

**Weaknesses:**

**1. Writing Clarity Issue: confusing concepts of “websites” and “domains”.**
> Different websites do not necessarily associate with in-domain and out-of-domain. Two websites can either belong to the same domain (e.g., both United and Spirit Airlines are for traveling) or not (e.g., Amazon for shopping and United for traveling). That being said, this paper aims to induce skills that transfer across websites, but it is unclear (at least from the intro) if these are websites within the same domain or across different domains.
Further, from the experiments, the proposed method only works for websites in-domain, so claiming its effectiveness in out-of-domain scenarios lacks evidence.

**2. Missing Skill Count Measures.**
> While the success rate and efficiency gains are promising, the paper fails to report the number of skills (abstract class and specialized implementations). Ensuring a small number of skills is important as it avoids redundancy in the skill library. The very low number of steps introduces concerns that there may be one skill unnecessarily targeting each specific example, instead of trying to capture shared similarities across examples. Reporting this dimension, similar to Table 3 in [1,] would help address this concern.

**3. Minor**
> but better remove the vspace before the “Conclusion” section title.

Once these missing aspects are added to the revised version, I am happy to increase the scores.

[1] Wang, Zhiruo, Daniel Fried, and Graham Neubig. "Trove: Inducing verifiable and efficient toolboxes for solving programmatic tasks." arXiv preprint arXiv:2401.12869 (2024).

**Questions:**

1. How different are the skill implementations within the same abstract class? Are they mainly different due to the varied designs of targeting websites, or other reasons? How similar (quantitatively) do two skill implementations need to be, in order to sit in the same abstract skill class?

---

> ### Author Response · Authors · 2025-11-23
> **Response to Reviewer M5TL**
>
> Thanks for your detailed review and constructive comments. We appreciate your recognition of the importance of skill transferability and found the experiments comprehensive to cover the actual use cases. Below, we provide detailed responses to each of your comments and hope to address any further considerations you may have.
>
>
> **W1: Terminology on "websites" vs. "domains"**
>
> Thank you for highlighting the distinction between "websites" and "domains." We agree that our method, PolySkill, is mainly effective in **cross-website** scenarios. This is confirmed by our primary results: Mind2Web Cross-website settings (Section 4.1), Continual Learning (Section 4.2), and Self-exploration (Section 4.3).
>
> While we retain cross-domain results for completeness, we acknowledge that our performance gains are less significant in this settings. Accordingly, we have revised the manuscript to strictly define our scope as **cross-website, in-domain transfer**. We have removed ambiguous language regarding out-of-domain generalization to ensure our claims align precisely with the experimental evidence.
>
> **W2: Skill Count Measures**
>
> Thanks for bringing up the issue of library size and generalization. We agree that reporting the number of learned skills is essential to verify that the agent is learning reusable abstractions rather than simply memorizing redundant paths.
>
> To demonstrate this, we measure the number of skills by measuring how many unique skills are learned in the Mind2Web settings with model GPT-4.1. We include both the *w/o Update* and *w/ Update* settings.
>
> | | **Cross-task** | | **Cross-Website** | | **Cross-Domain** | |
> | :--- | :---: | :---: | :---: | :---: | :---: | :---: |
> | **Method** | **Acc** | **# Skill**  | **Acc** | **# Skill** | **Acc** | **# Skill** |
> | *Baseline* | 53.8 | - | 56.2 | - | 62.3 | - |
> | | | | | | | |
> | *Static Approaches* | | | | | | |
> | ASI | 52.3 | 50 | 54.9 | 47 | 57.3 | 33 |
> | **PolySkill** | 55.4 | 43 | 57.6 | 44 | 60.1 | 36 |
> | | | | | | | |
> | *w/ Online Update* | | | | | | |
> | ASI + Update | 59.4 | 66 | 58.7 | 71 | 62.1 | 66 |
> | **PolySkill + Update** | **63.2** | **47** | **61.3** | **53** | **63.4** | **46** |
>
> From the table, we can see that PolySkill consistently achieves higher accuracy with a more compact skill library. Notably, in the Cross-task (Online) setting, PolySkill improves performance by nearly 4% over ASI while reducing the library size by ~29% (47 vs. 66 skills). This suggests that PolySkill successfully distills versatile, polymorphic abstractions that adapt to new contexts, whereas baselines tend to accumulate redundant, environment-specific skills to handle variations. We have updated the new results and corresponding analyses under Section 4.2 in the updated manuscript.
>
>
> **Q1: Difference on the skill implementations**
>
> Great question. Skill implementations within the same abstract class differ primarily due to website-specific designs: each site has its own A11y tree IDs, and layout patterns. For example, two `add_to_cart` implementations might involve completely different action sequences depending on whether a site uses a dropdown menu, a modal dialog, or a direct button click.
>
> However, despite these implementation differences, they share the same abstract function signature and logical purpose, which is what allows them to sit in the same abstract class. This separation between abstract function and concrete implementation is exactly what makes compositional skills powerful. The agent only need to know what a skill does from the abstraction class, not how it's implemented on each specific site. Quantitatively, we found that function overlap between shopping skill implementations in Section 4.3, ranges from a minimum of 3.2% to a mean of around 35.7%. This shows that even with relatively low code overlap, skills can belong to the same abstract class as long as they serve the same logical function.
>
> The threshold isn't rigid; what matters is whether skills accomplish the same high-level goal. This abstraction is particularly powerful for building compositional skills. Because the interface is standardized, a high-level skill (like buy_now) can simply call the abstract add_to_cart function. It does not need to know if the underlying concrete implementation requires a modal dialog or a direct click—it just works.
>
> We'll clarify this in the revised manuscript to make the abstraction criteria clearer.
>
>
> **W3: Minor formatting**
>
> Thank you for pointing out the format issue. We have removed the `vspace` command as requested in the updated manuscript.

---

### Official Review · Reviewer_dJWv · 2025-10-27

**Soundness:** 2
**Presentation:** 3
**Contribution:** 2
**Rating:** 4
**Confidence:** 3

**Summary:**

The paper presents a framework in which an LLM agent learns skills that can be reused across websites (e.g., "search", "add_to_cart", "checkout"). At the core of this framework is the separation of "what" from "how" (e.g., the framework defines an abstract shopping website class that is specialized to different actual shopping websites). Experiments compare the performance of the proposed system with prior work on existing benchmarks.

**Strengths:**

- The paper clearly positions the proposed apprpoach relative to prior work (ASI), making it easy to understand the technical contribution.
- The paper cleverly uses polymorphism, a well-understood concept from classic software engineering principles, to improve generalizability in LLM agents (although I am not an expert in this area, and thus I cannot be certain of my assessment of novelty).

**Weaknesses:**

- The abstract class that specifies the relevant skills is provided as context. This is a very restrictive assumption, as it assumes (1) that the set of useful skills is fixed, (2) that the set of useful skills is known.
- I found the current description to be insufficient to understand the method, and had to consult the appendix. Please consider improving the framework's pseudocode to the main text, or at least including a precise step-by-step description in the main text.
- It seems that none of the baseline systems were given the extra information corresponding to the abstract class description (or this was at least not described), which makes the comparison perhaps biased. Giving this information to at least some of the systems should be possible (perhaps simply by including it in the prompt).
- Some parts of the methodology are underdocumented (see questions 1, 2, 3, 5).

**Questions:**

1. Can the abstract class definition be inferred from the initial successful traces?
2. Can the system accidentally use a not-yet-implemented skill in its implementation of a skill?
3. How are the "+Update" systems implemented?
4. Please consider running the experiments providing the information corresponding to the abstract class to the baseline systems that allow it. This would further allow us to understand if the performance benefits come simply from the apriori identification of relevant skills or actually from the polymorphic skills.
5. From the manuscript, I am guessing the +Update and +Online systems modify their implementation of the concrete classes. Is this corrrect? Or do these systems also update the weights of the LLMs? Please include a description of these systems in the main text.

---

> ### Author Response · Authors · 2025-11-23
> **Response to Reviewer dJWv**
>
> We thank the reviewer for the detailed review and constructive comments. We appreciate your recognition of the clear writing and the novelty of applying polymorphism to LLM agents. Below, we address your specific concerns and clarify the misunderstandings regarding the origin of the abstract class.
>
> **W1 & W3 & Q1: Restrictive assumption on the abstract class as context and unfair baseline comparison**
>
> We clarify that the impression that the abstract class is provided as external human knowledge likely stems from the illustrative example in Table 1. This is not the case. We want to clarify that the initial abstract class definition contains only the class name (e.g., `AbstractShoppingSite`) and a generic docstring. All skills, including both the abstract signatures and the site-specific implementations, are automatically induced by the agent during the induction process. Therefore, the restrictive assumptions mentioned do not hold in our framework.
>
> First, the set of useful skills is not fixed; it continues to evolve and grow during inference time as new tasks are encountered. Second, the set of skills is not known in advance or manually defined; the agent discovers necessary skills (like `add_to_cart`) based on its exploration. Because this "extra information" is generated by the agent itself from the first website it visits rather than provided by humans, PolySkill does not have an unfair advantage over the baselines. Both start with the same information; PolySkill simply leverages the experience on the first site to build a reusable schema. We have updated the Methodology section to explicitly clarify this induction process.
>
> **W2: Improving the framework's pseudocode in the main text**
>
> Thank you for the suggestion. As noted above, we have moved the framework's pseudocode to the main text to ensure the method is self-contained and easier to follow.
>
> **Q2: Can the system use a not-yet-implemented skill?**
>
> This is a valid concern. It is rare, but possible, for the agent to call a skill that is abstractly defined but lacks a site-specific concrete implementation. This typically happens when the agent visits a new website and the planner attempts to use a known abstract skill (e.g., `checkout`) before the induction module has created the code for that specific URL. In these cases, the environment returns an execution error. The agent is then given another turn to recover, usually by attempting an alternative primitive action or triggering the exploration module. We observe this primarily in self-exploration settings, and the agents are generally capable of recovering from these errors effectively.
>
> **Q3 & Q5: How are +Update and +Online actually implemented?**
>
> First, we clarify that **+Update** and **+Online** refer to the exact same experimental setting in our initial submission. To prevent future confusion, we have unified these under the single term **+Update** throughout the revision. These variants refer to updating the skill library (the code available in the prompt context) at test time by performing the skill induction process. We do not perform any gradient updates (weight training) on the LLM itself. This setup simulates how a tool library evolves in the wild, where the agent accumulates verified concrete skill implementations for newly encountered environments. We have moved the detailed pseudocode from the Appendix to the main text (Section 3) and improved the description to clearly distinguish between library updates (context) and parameter updates (weights).

---

> > ### Comment · Reviewer_dJWv · 2025-11-25
> >
> > Thank you for your response and for updating the manuscript. I have revised my score.

---

### Official Review · Reviewer_kozN · 2025-11-01

**Soundness:** 2
**Presentation:** 2
**Contribution:** 2
**Rating:** 4
**Confidence:** 4

**Summary:**

The paper proposes PolySkill, a polymorphism-guided framework for skill induction in web agents. A skill is defined by an abstract interface capturing semantic intent, linked to multiple interchangeable concrete implementations across websites. This separation aims to improve transfer, modularity, and recomposition of skills, and to support continual learning and adaptive behavior in task-free settings. The authors also introduce process-level metrics—Skill Reusability, Task Coverage, and Skill Compositionality—to assess what the agent actually learns beyond task success.

**Strengths:**

1. Skills are specified by an abstract interface plus interchangeable concrete implementations, enabling cross-site reuse and composition while insulating from UI changes.
2. A three-stage flow consisting of polymorphic abstraction, compositional verification, and adaptive execution clarifies how skills are discovered, validated, and deployed.

**Weaknesses:**

1. It’s unclear how much gain comes from the interface abstraction vs. better verification, task curricula, retrieval, or engineering heuristics.
2. Most compelling examples are shopping; results on other domains (dev tools) are thinner and may conflate prior knowledge with schema effects.

**Questions:**

1. How is a domain’s abstract class discovered, manually seeded, or fully induced? What prevents over-abstracting or under-abstracting?
2. How is the $\gamma$ penalty on steps set; is it tuned per model/domain; sensitivity?
3. What is the compute & API cost of induction/verification vs. Baselines?

---

> ### Author Response · Authors · 2025-11-23
> **Response to Reviewer kozN [1/2]**
>
> We thank the reviewer for the detailed review and constructive comments. We appreciate your recognition of our skill induction pipeline and are glad you found the experiments compelling. Below, we provide detailed responses to your specific questions and concerns.
>
> **W1: Unclear on the gain with PolySkill (Attribution of performance)**
>
> Thank you for raising this concern regarding fairness. We want to emphasize that **we utilized the exact same verification (LLM-as-Judge) and retrieval settings** for both the baselines (ASI and SkillWeaver) and our method. Therefore, the performance gains observed in PolySkill cannot be attributed to superior verification or retrieval engineering.
>
> Instead, the gain stems directly from the **polymorphic structure**. By separating shared abstract behaviors from site-specific implementation details, we reduce the risk of "context rot" described in recent literature [1,2]. In the baselines with transfer settings, the context window often becomes cluttered with irrelevant specific actions from other websites, leading the model to hallucinate or select incorrect skills. PolySkill’s hierarchical structure ensures the agent only accesses the relevant concrete implementation for the current site, thereby cleaning the context window and reducing selection errors.
>
> [1] Hong, K., Troynikov, A. and Huber, J. (2025) Context Rot: How Increasing Input Tokens Impacts LLM Performance. Technical Report. Chroma, July. Available at: https://research.trychroma.com/context-rot.
>
> [2] Anthropic. "Effective Context Engineering for AI Agents." (2025), https://www.anthropic.com/engineering/effective-context-engineering-for-ai-agents.
>
> **W2: Domain breadth and limitation**
>
> We acknowledge that our experiments focused primarily on depth within complex shopping and developer sites rather than broad domain coverage. We selected Shopping as our primary case study because it serves as the standard running example in the field, notably in the ASI benchmark [3]. While our method is domain-agnostic and can be extended to flight booking, social media, or administrative workflows, we prioritized a deep analysis of cross-site transfer within the shopping domain for this study.
>
> We have added a **Cost Analysis** (detailed in Q3 below) to the Appendix, noting that the high compute cost of agentic evaluation was a factor in limiting the number of additional domains in this iteration. We have explicitly stated this scope limitation in the Conclusion.
>
> [3] Wang, Zora Zhiruo, et al. "Inducing programmatic skills for agentic tasks." arXiv preprint arXiv:2504.06821 (2025).

---

> ### Author Response · Authors · 2025-11-23
> **Response to Reviewer kozN [2/2]**
>
> **Q1: Discovery of domain abstract class**
>
> In our frameworkour framework, the abstract class is purely induced by LLMs. Our contribution is the *induction* of the concrete implementations for each website and the polymorphic framework that links them. We have clarified this distinction in Section 3 to be more upfront about this assumption. The control of abstraction are based on prompting only, but as mentioned in the future work, with training model to enable them to pick up skills, it can be controlling the level of abstraction by manipulating the $\gamma$ penalty.
>
> **Q2: $\gamma$ Step penalty sensitivity**
>
> Currently, the step penalty ($\gamma$) is a **fixed heuristic** imposed during the inference/planning stage. We do not tune it per domain in this iteration. Since we are not currently training the model weights , we cannot incorporate this penalty directly into a loss function during induction. We identify that $\gamma$ can be used for reward shaping in Reinforcement Learning, which is a promising direction for future work to dynamically control the trade-off between exploration steps and skill usage.
>
> **Q3: Compute & API cost analysis**
>
> Thanks for bringing up the critical issue of efficiency. We have conducted a detailed cost evaluation measuring both the steps taken and token consumption.
>
> The table below details the average token consumption (input + output) per instance in the Mind2Web cross-task settings:
>
> | Method | Success Rate | Action Gen. | Verification | Skill Induction | Total Tokens |
> | :--- | :---: | :---: | :---: | :---: | :---: |
> | Baseline (No Skills)| 53.8 | 55,367.4 | -- | -- | 55,367.4 |
> | ASI | 59.4 | 53,840.5 | 4,673.8 | 5,773.2 | 64,287.5 |
> | **PolySkill** | **63.2** | **49,710.3** | 4,963.2 | 8,476.4 | **63,149.9** |
>
> **Key Findings:**
> 1.  **Lower Inference Cost:** PolySkill significantly reduces token consumption during the **Action Generation** phase (~49.7k vs ~55.4k baseline). The induced polymorphic skills allow the agent to solve tasks in fewer steps, reducing total context processing.
> 2.  **Amortized Efficiency:** While the **Skill Induction** phase introduces overhead, the Total Token consumption is actually lower than the ASI baseline (63.1k vs 64.3k), while achieving higher success rates (63.2% vs 59.4%).
> 3.  **Long-term Savings:** Crucially, verification and induction are primarily **one-time costs**. Once a concrete skill is induced for a website, it is reused for all future tasks on that site. This makes PolySkill highly attractive for deployment, as the recurring inference cost is **reduced by ~10.2% compared to the Baseline** and **~7.7% compared to ASI**.
>
> We have updated these results and the discussion in **Appendix F.1** of the revised manuscript.

---

### Official Review · Reviewer_hnAi · 2025-11-02

**Soundness:** 4
**Presentation:** 4
**Contribution:** 3
**Rating:** 8
**Confidence:** 4

**Summary:**

This paper proposes a method that improves the reusability of skill induction methods for agents by encouraging the agent to create abstractable skills using polymorphism.

**Strengths:**

1. The method is simple but strikes at the core of the problem.
2. The motivation and description is very clear, illustrated by appropriate examples.
3. The paper compares across multiple datasets against strong baselines from the recent literature.

**Weaknesses:**

1. As mentioned in the future work section, it is less clear how this method will be applicable in cases where task category boundaries are more fuzzy. The hard-coded task boundaries required to induce new abstract classes may make this tricky. I don't view this as a critical weakness of the paper though, more of an opportunity for future work.

**Questions:**

One thing that was not very clear to me (sorry if I missed this) -- is the abstract class only created once when the web site is first processed? Or can it be modified later as more evidence about the web site comes to light?

---

> ### Author Response · Authors · 2025-11-23
> **Response to Reviewer hnAi**
>
> We thank the reviewer for the positive assessment, particularly for highlighting the simplicity of our method and the clarity of our motivation. We appreciate your constructive feedback and address the specific concerns below.
>
> **W1: Applicability in cases where task category boundaries are fuzzy.**
>
> **A:** We appreciate this insightful observation. We agree that the current reliance on manually defined boundaries is a necessary scope designed to stabilize the initial polymorphism framework. As you noted, extending this to the general case is a prime candidate for future work. We envision addressing this by moving from manual boundaries to **autonomous skill clustering**, where the agent proposes abstract classes based on functional similarity (e.g., via soft-clustering of execution traces [1]) rather than predefined labels. This would allow "abstract classes" to emerge organically. We will include a discussion on this potential approach in the `Future Work` section.
>
> **Q1: Is the abstract class created once, or can it be modified later?**
>
> **A:** In our current framework, the **abstract class structure** (e.g., `AbstractShoppingSite`) is initialized at the beginning of training. However, the **set of abstract skills** (functions such as `add_to_cart`) and compositional skills are continuously updated as new websites and tasks are encountered. Similarly, the **concrete implementations** for each website that inherit from the abstract class (e.g., `AmazonShopping`) are dynamic. As the agent explores a website with new tasks, it induces, verifies, and updates the concrete implementation of the relevant abstract skills for that specific site.
>
> We have included the clarification: **"an evolving abstract interface and a dynamically growing library of concrete implementations"** in the updated manuscript under the method section. As you rightly note, adapting the abstract class itself to better handle "fuzzy boundaries" is a very exciting direction for future work.
>
> [1] Peng, Shaohui, et al. "Self-driven grounding: Large language model agents with automatical language-aligned skill learning." arXiv preprint arXiv:2309.01352 (2023).

---

> > ### Comment · Reviewer_hnAi · 2025-11-25
> > **Thank you for the response**
> >
> > Thank you for the response, I still think this is a nice paper and think it deserves to be accepted.

---

### Author Response · Authors · 2025-11-23
**General Response to All Reviewers [1/2]**

We thank all four reviewers for their time and their detailed, constructive feedback. We are very encouraged that the reviewers converged on several key strengths of our work.

* **Novel Idea based on Polymorphism:** Reviewers consistently praised the novelty of grounding agentic skill learning in **polymorphism**, a classic software engineering principle. They noted that this "clever" (**dJWv**) application "strikes at the core of the problem" (**hnAi**) by separating the abstract interface from concrete implementations, which enables cross-site reuse and insulates agents from brittle UI changes (**kozN**, **M5TL**).
* **Comprehensive Experiments:** Our evaluation was recognized as **comprehensive and rigorous**. Reviewers appreciated that we tested across multiple datasets against strong baselines (**hnAi**) and highlighted the empirical improvements shown in both static benchmarks and open-ended exploration settings (**M5TL**). Reviewer **kozN** specifically found the experiments "compelling," validating our claims on transferability and efficiency.
* **Important Topic:** Reviewers agreed that addressing **transferability across websites** is a critical limitation of current studies (**M5TL**) and that our focus on reusability addresses a significant open problem in the field (**hnAi**, **kozN**).

We appreciate this positive feedback and will now address the primary common concern raised regarding the induction process.




### **Common Clarification: The Origin of Abstract Classes & Fairness**

We thank reviewers **dJWv** and **kozN** for raising crucial questions regarding how the abstract class is defined. Reviewer **dJWv** noted this might be a "restrictive assumption" if provided manually, and **kozN** asked if it is "manually seeded or fully induced."

**We clarify that the abstract class is NOT external information provided a priori by humans.**

In the original manuscript, the mechanism for creating the abstract class (e.g., `AbstractShoppingSite`) was under-explained, leading to the impression that it was a manual "hint" given to the model. We have updated the **Methodology** section to explicitly state:

1.  **Fully Induced:** The abstract class is **automatically induced** by the agent during its interaction with the first website it encounters in a category (e.g., Amazon). The agent analyzes successful traces to define the function signatures (the "what") before defining the concrete implementation (the "how").
2.  **No Unfair Advantage:** Because this "extra information" (the abstract class) is generated by the agent itself from the training data rather than provided by humans, **PolySkill does not have an unfair advantage over the baselines.** Both our method and the baselines (ASI, SkillWeaver) start with the same information. PolySkill simply leverages that initial experience to build a reusable schema, whereas baselines treat the experience as isolated.
3.  **Dynamic Evolution:** As noted in our response to **hnAi**, the set of useful skills is **not fixed** and **not known in advance**. The agent continues to discover and add new abstract skills (like `add_to_wishlist`) as it explores new tasks.

---

> ### Author Response · Authors · 2025-11-23
> **Summary of the Update and New Revision of our Paper [2/2]**
>
> Based on the feedback, we have updated our manuscripts accordingly and uploaded a new version of our paper for review. The changes are highlighted in blue. We summarize the key changes:
> * **Revised the scope definition** in the **Introduction**, strictly defining the focus as "cross-website, in-domain transfer" to distinguish from broad cross-domain generalization (Reviewer M5TL).
>
> * **Clarified the origin of the abstract class** in **Section 3**, explicitly stating that it is automatically induced from the first website encounter rather than provided as a manual prior (Reviewer dJWv).
>
> * **Moved the core framework pseudocode** from the Appendix to **Section 3** to ensure the methodology is self-contained (Reviewer dJWv).
>
> * **Clarified the +Update and +Online mechanisms** in **Section 3**, specifying that they rely on in-context library updates rather than gradient-based parameter updates (Reviewer dJWv).
>
> * **Added a distinction between abstract and concrete evolution** in **Section 3**, clarifying that the abstract interface grows while concrete implementations are dynamic and site-specific (Reviewer hnAi).
>
> * **Added a new Library Size analysis** in **Section 4.2**, including a new table comparing Accuracy vs. Number of Skills to demonstrate PolySkill's efficiency (Reviewer M5TL).
>
> * **Incorporated experimental findings** showing performance when the abstract definition is entirely self-induced (inferred) in **Section 4** (Reviewer dJWv).
>
> * **Added Future Work section** to include **autonomous skill clustering** (soft-clustering for fuzzy boundaries) and **Reinforcement Learning** (using $\gamma$ for reward shaping), as requested by Reviewers hnAi and kozN.
>
> * **Explicitly acknowledged domain breadth limitations** in the **Conclusion**, explaining the focus on depth over breadth due to compute constraints (Reviewer kozN).
>
> * **Added a detailed Cost & Efficiency Analysis** in **Appendix F.1**, including a table breaking down token consumption for Action Generation vs. Induction to demonstrate amortized efficiency (Reviewer kozN).
>
> * **Removed `vspace` commands** throughout the manuscript to comply with standard formatting requirements (Reviewer M5TL).

---

### Author Response · Authors · 2025-12-01
**Note to AC (summary for rebuttal phase)**

Dear Area Chair,

We note the recent information leak incident on OpenReview and sincerely appreciate your time and effort in reviewing our paper. In our final comment, we would like to provide a summary of the rebuttal phase and clarify the status of our reviewer interactions.

* **All four reviewers found the problem formulation and results to be important:**
  * **hnAi:** The method "strikes at the core of the problem" of reusability and noted the work "deserves to be accepted."
  * **dJWv:** The work "cleverly uses polymorphism... to improve generalizability" in LLM agents and makes it "easy to understand the technical contribution."
  * **M5TL:** The paper "tackles a critical limitation of current agent skill learning studies – transferability across websites" and demonstrates "empirical improvements throughout experiments."
  * **kozN:** The approach enables "cross-site reuse and composition while insulating from UI changes" and "clarifies how skills are discovered, validated, and deployed."

* **We made changes to improve the paper. In the rebuttal period, we made the following changes to the paper**:
  * **Clarified the Origin of Abstract Classes.** Reviewer dJWv and kozN raised concerns that the abstract class might be a manual prior. This is incorrect, and we've updated the methodology to explicitly state that the abstract class is **fully automatically induced** by the agent from the first website encounter, **ensuring no unfair advantage over baselines**.
  * **New Cost and Efficiency Analysis.** Reviewer kozN asked for the compute cost of induction vs. baselines. We added a detailed analysis (Appendix F.1) showing that while induction adds overhead, PolySkill reduces inference token consumption by ~10.2% and total tokens by ~7.7% compared to baselines.
  * **Library Size Analysis and Scope Refinement.** Reviewer M5TL requested metrics on the number of skills to prove we are not just memorizing. We added a new table (Section 4.2) showing PolySkill achieves higher accuracy with a **smaller, more efficient skill library** (47 skills vs 66 for baselines). We also moved the core pseudocode to the main text to improve clarity as requested by dJWv.

* **The reviewer who acknowledged reading our rebuttal raised their score.**
  * **Reviewer dJWv** (Score: 4 $\to$ 6) stated that their main concern was the "restrictive assumption" that the abstract class was provided as context.
    * **Response:** We clarified that the abstract class is fully induced. The reviewer acknowledged this, stating "Thank you for your response and for updating the manuscript. I have revised my score".
    * **Status:** The reviewer raised their score from a **4 $\to$ 6**. However, due to the system rollback, this score update was reverted.

* **We believe we have addressed the main concerns of the remaining reviewers.**

  * **Reviewer hnAi (Score: 8)** maintained a strong positive score throughout.
    * **Concern:** Their only minor concern was regarding "fuzzy boundaries" for task categories.
    * **Response:** We clarified that handling fuzzy boundaries via autonomous skill clustering is a key direction for future work. The reviewer replied, "Thank you for the response, I still think this is a nice paper and think it deserves to be accepted".

  * **Reviewer kozN (Score: 4)** found the experiments "compelling" but asked about the "fairness" of the verification setup and "compute costs".
    * **Response:** We confirmed that we use the exact same verification/retrieval as baselines (isolating the gain to our method) and provided the requested cost analysis proving efficiency. We believe these additions directly addressed the "fairness" and "cost" questions that were central to their score.

  * **Reviewer M5TL (Score: 6)** explicitly stated, "Once these missing aspects are added to the revised version, I am happy to increase the scores".
    * **Concern:** The missing aspects were 1) clarity on "websites vs domains" and 2) "missing skill count measures".
    * **Response:** We revised the manuscript to strictly define the scope as "cross-website" and added the requested skill count analysis in Section 4.2. Since we directly provided the specific data requested for the score increase, we are optimistic that the reviewer would have raised their score had the discussion continued.

If at all possible, we would greatly appreciate it if the AC could take into consideration that the reviewer who was able to engage (dJWv, hnAi) found their concerns addressed and raised their score. We also believe that we have largely addressed the concerns of the reviewers who did not engage in discussion. We believe that the reviewers all found the problem we are studying to be important and of interest, and believe that we have significantly addressed the stated concerns.

Best regards,

The Authors

---

### Meta-Review · Area_Chair_CaFA · 2026-01-06

**Summary:**

The paper addresses an important limitation of current agent skill learning, namely reusability and transferability across websites. The proposed polymorphism-guided formulation is intuitive and practically motivated, and the empirical results show consistent gains over strong baselines across both static benchmarks and continual learning settings.

Reviewers generally found the problem important and the idea promising, with empirical gains suggesting improved transfer and efficiency. The main concerns focused on evaluation fairness and assumptions, and on method clarity/implementation details. In addition to these issues, the paper’s writing quality and technical presentation were noted to be weak in places, with confusing or error-prone explanations that made key mechanics harder to verify from the original submission.

Reviewer concerns were largely addressed in the rebuttal, especially the fairness-related assumption about whether abstract classes are manually provided. The authors clarified that abstract interfaces and skill signatures are induced by the agent rather than given as external priors, and they improved the main-text description by adding pseudocode and additional analyses on cost and skill library size. At least one reviewer explicitly acknowledged these clarifications and revised their score upward.

I consider this paper to be borderline acceptable due to substantial remaining presentation issues. Beyond the concerns raised by reviewers, the manuscript contains multiple incorrect internal references (tables/figures pointing to the wrong items), which makes it harder to verify claims and follow the method. More importantly, the new pseudocode introduced during the rebuttal appears to contain serious factual errors in describing the method’s mechanics (e.g., the formulas related to the skill library ​​K update in Lines 9 and 11 are incorrect), which raises concerns about clarity and reproducibility even if the underlying idea is sound.

**Reviewer Concerns:**

**Addressed concerns:**

- Fairness of comparison and origin of abstract classes (dJWv, kozN): Reviewers questioned whether the abstract class was manually provided and whether this gave PolySkill an unfair advantage over baselines. The authors clarified that the abstract class and skill signatures are fully induced by the agent from its first website encounter, and that baselines do not receive extra information. Reviewer dJWv acknowledged this clarification and revised their score.

- Method clarity and algorithmic details (dJWv): Reviewers requested clearer descriptions of the framework, including moving pseudocode to the main paper and clarifying the meaning of +Update and +Online. The authors moved the core pseudocode to Section 3 and clarified that these variants correspond to skill library updates rather than model weight updates.

- Compute cost and efficiency (kozN): Reviewers asked for a breakdown of induction and verification costs relative to baselines. The authors added a detailed cost and token consumption analysis showing amortized efficiency compared to ASI.

**Outstanding concerns:**

- Attribution of gains to polymorphic abstraction (kozN): While the authors argued that performance gains stem from the polymorphic structure rather than verification or retrieval choices, the reviewer’s concern about fully isolating this factor was not conclusively resolved.

- Breadth of evaluation across domains (kozN, M5TL): Reviewers noted that most experiments focus on shopping-style websites, and evidence for broader domain coverage remains limited.

**Reviewer Scores:**

**hnAi (8):** Likely unchanged. The reviewer maintained a strong positive assessment throughout the discussion. After the rebuttal, the reviewer explicitly stated that they still believe the paper deserves acceptance.

**dJWv (4 to 6):** Likely increased. The reviewer’s main concern was the assumption that the abstract class was provided as a manual prior. After the authors clarified that the abstract class is fully induced and revised the manuscript accordingly, the reviewer acknowledged the clarification and revised their score upward. The score update was reverted due to a system rollback.

**M5TL (6):** Likely unchanged or slightly increased. The reviewer explicitly stated willingness to increase the score once the missing clarifications on scope and skill count were added. The authors provided the requested revisions, but the reviewer did not post a follow-up response.

**kozN (4):** Likely unchanged. The reviewer noted that most experiments focus on shopping-style websites; however, the authors did not provide convincing evidence in the rebuttal for the method’s broader applicability.

---

### Decision · Program_Chairs · 2026-01-26

Accept (Poster)